# Intracortical recordings reveal vision-to-action cortical gradients driving human exogenous attention

Tal Seidel Malkinson [1,2] ✉, Dimitri J. Bayle [3], Brigitte C. Kaufmann[1], Jianghao Liu [1,4], Alexia Bourgeois[5], Katia Lehongre [6], Sara Fernandez-Vidal[6], Vincent Navarro[1,7,8], Virginie Lambrecq[1,7,8], Claude Adam[1,7,8], Daniel S. Margulies [9], Jacobo D. Sitt[1] & Paolo Bartolomeo [1]

Exogenous attention, the process that makes external salient stimuli pop-out of a visual scene, is essential for survival. How attention-capturing events modulate human brain processing remains unclear. Here we show how the psychological construct of exogenous attention gradually emerges over large-scale gradients in the human cortex, by analyzing activity from 1,403 intra-cortical contacts implanted in 28 individuals, while they performed an exo-genous attention task. The timing, location and task-relevance of attentional events defined a spatiotemporal gradient of three neural clusters, which mapped onto cortical gradients and presented a hierarchy of timescales. Visual attributes modulated neural activity at one end of the gradient, while at the other end it reflected the upcoming response timing, with attentional effects occurring at the intersection of visual and response signals. These findings challenge multi-step models of attention, and suggest that frontoparietal networks, which process sequential stimuli as separate events sharing the same location, drive exogenous attention phenomena such as inhibition of return.

Imagine sitting in your car, waiting for the traffic light to change, when suddenly an adjacent billboard sign starts flashing, capturing your attention. How would the flashing sign affect your ability to subse-quently detect the light changing to green? In such a situation, the flashing automatically renders the sign more salient in the visual scene through a fast and dynamic orientation process known as exogenous attention. Exogenous attention is a fundamental process that mod-ulates response speed and perceptual sensitivity[1] and is prevalent among many vertebrate species[2–4], yet the expansion of attention systems in the human brain sets us apart[5]. Understanding how our

brain handles such salient distractions has become ever more crucial in our information-saturated modern environment. Yet, what exactly determines if our attention will be captured or reoriented away is not clear. Attention's temporal dimension, that is, how a previous stimulus, such as a salient attention-capturing cue, affects the processing of a subsequent stimulus, such as a target, is a key element for answering this important question. For instance, when successive stimuli appear at the same location within short delays, they lead to faster perfor-mance (response time (RT) facilitation). Slightly longer delays, how-ever, slow down responses, a phenomenon termed inhibition of return

[1]Sorbonne Université, Inserm UMRS 1127, CNRS UMR 7225, Paris Brain Institute, ICM, Hôpital de la Pitié-Salpêtrière, 75013 Paris, France. [2]Université de Lorraine, CNRS, IMoPA, F-54000, Nancy, France. [3]Licae Lab, Université Paris Ouest-La Défense, 92000 Nanterre, France. [4]Dassault Systèmes, Vélizy-Villacoublay, France. [5]Laboratory of Cognitive Neurorehabilitation, Faculty of Medicine, University of Geneva, 1206 Geneva, Switzerland. [6]CENIR - Centre de Neuro-Imagerie de Recherche, Paris Brain Institute, ICM, Hôpital de la Pitié-Salpêtrière, 75013 Paris, France. [7]AP-HP, Epilepsy and EEG Units, Pitié-Salpêtrière Hospital, 75013 Paris, France. [8]Reference center of rare epilepsies, EpiCare, Pitié-Salpêtrière Hospital, 75013 Paris, France. [9]Laboratoire INCC, équipe Perception, Action, Cognition, Université de Paris, 75005 Paris, France. ✉e-mail: tal.seidel-malkinson@univ-lorraine.fr

(IOR), which may promote spatial exploration[6,7]. Under certain conditions (e.g., when cue and target do not overlap in time), IOR may even offset RT facilitation[8]. These opposing RT modulations reflect underlying attentional processes[9]. However, despite decades of research, the nature and underlying neural mechanisms that mediate these attentional effects remain unclear[10,11]. Evidence from human and non-human studies suggests that information about physical salience, which guides exogenous attention, may emerge as early as the primary visual cortex, but this is still debated[12,13]. There are mixed results about the brain localization of such activities and about the specific stimulus features that elicit exogenous attention[14–16]. Salience information converges with top-down influences in several higher-order areas related to attention[13,17,18]. In humans, attention-related networks include a dorsal frontoparietal network and a more right-lateralized ventral network, comprising the temporoparietal junction (TPJ) and the ventral prefrontal cortex[19]. Global salience may be computed within salience maps in the parietal cortex[18,20–22] or the prefrontal cortex[22–24], as well as in subcortical structures such as the superior colliculi and the pulvinar[25]. Several of these areas, such as the superior colliculi, the frontal eye fields (FEF), the posterior parietal cortex, and their connections, were also shown to be involved in IOR[26–34]. For example, dysfunction of these regions in the right hemisphere[35] causes spatial neglect, a condition characterized by a failure to orient attention to left-sided events and persistent RT facilitation instead of the typical IOR for right-sided targets[33,34], linking abnormal exogenous attention to this disabling neurological condition. However, there is no consensus regarding the exact nature and neural basis of IOR[10,36] and very little effort was directed into exploring the neural basis of RT facilitation, with no single neural marker of these effects identified[11]. There are several contentious neural theories of IOR, but very few about RT facilitation, and the evidence supporting each of them is limited, indirect, and often contradictory. Theories of IOR diverge on the mechanistic nature of IOR and its putative localization(s) in the brain (sensory/attentional and/or motor/decisional). It was suggested, for instance, that IOR is caused by attentional capture of previously cued locations[37], perhaps by delaying bottom-up signals of the salience map[10,26,27,38], or by an inhibitory attentional bias[39,40]. A recent theoretical model[41] based on the known architecture of frontoparietal cortical networks and on their anatomical and functional asymmetries[42] proposed that IOR arises from a noise-increasing reverberation of activity within priority maps of the frontoparietal circuit linking frontal eye field (FEF) and intraparietal sulcus (IPS). Other theories proposed that IOR might occur early, over perceptual neural pathways through the reduction of stimulus salience around a previously attended location[43], or due to sensory adaptation[44] or habituation[45]. IOR was suggested to occur also later in processing, involving motor/decision circuits, in the form of a bias against responses toward previously attended spatial locations[43], motor habituation[45] or an oculomotor activation signal[46]. For example, the cue-target event integration-segregation hypothesis[6] postulates that the summation of early and late perceptual processes, spatial selection processes, and decision processes determines together if the net behavioral effect is facilitatory (RT facilitation) or inhibitory (IOR)[6,11,36]. According to this theory, binding together of sequential stimuli that share similar features (such as location and close-timing) into a single event file[47] can lead to facilitatory effects helping to select the target location in advance[6]. However, binding can also cause inhibitory effects when a similar sequential stimulus needs to be detected as a new separate event, resulting in a cost in detecting the onset of the target[6]. These theories remain highly debated, and the evidence supporting each one is inconclusive. This is due at least in part to the fact that prior work investigating the neural basis of these fast and dynamic processes is quite sparse and based either on high-resolution recordings in specific brain regions in non-human primates or on indirect human neuroimaging methods with limited spatial resolution, such as EEG, or with limited temporal resolution, such as functional MRI. These considerations are critical when studying the neural correlates of exogenous attention, which operates on a very rapid time scale and dynamically involves large neural networks over the entire brain, thus rendering past findings not informative enough to support or refute existing neural theories of attention. Thus, our understanding of these attention processes stays fragmented, leaving the involved networks and underlying mechanisms obscure.

Here we set out to establish the large-scale spatiotemporal neural dynamics of the mechanisms involved in the exogenous orienting of spatial attention. We chose to use intracortical EEG (iEEG) in humans[48–50], acquired across 28 patients (1403 contacts), to achieve comprehensive cortical coverage. iEEG is the only method that allows the tracking of human attentional dynamics directly (i.e., invasively) with high temporal resolution and excellent spatial precision over large brain topographies, crucial for capturing rapid attentional dynamics across the brain. Because of the lack of consensus on the neural basis of exogenous attention, we chose to use a data-driven approach, leveraging the advantages of iEEG to establish how neural activity tracks visual, attentional, and response aspects of the classic Posner exogenous attention task[7] and test whether the findings converge with existing theoretical frameworks. This approach allowed us to study the impact of attentional cues on the detection of subsequent targets as a function of the delay between them. Typically, depending on the congruence between cue and target locations and the cue-target delay, this task generates differences in RT (RT facilitation or IOR)[7,8]. We assumed that the activity of putative neural mechanisms underlying this exogenous attention RT effects should present: (1) visual spatial sensitivity; (2) sensitivity to cue-target delay; (3) sensitivity to task relevance (cue/target); (4) association with RT.

To study how the evoked activity relates to large-scale brain organization, we examined its mapping across the cortical gradient, an axis of variance in anatomical, functional, neurodevelopmental and evolutionary features, along which areas fall in a spatially continuous order[51–54]. The cortical gradient is a recently discovered organizing principle of cortical topography[51,53], based on the differentiation of connectivity patterns that captures a spatial and functional spectrum from early regions dedicated to perception and action (Periphery) to high-level regions of more abstract cognitive functions (Core)[53], akin to Mesulam's[55] unimodal-to-transmodal cortical hierarchy. Therefore, localizing activity along this gradient indicates the microstructural and genetic features, connectivity profile, and functional role of the activated region[51–53].

This combined approach sought to clarify the theoretical debate on the neural basis of exogenous attention by tracking precisely its neural correlates and mapping them onto the large-scale topography of the brain.

## Results

Twenty-eight participants undergoing presurgical evaluation of their epilepsy with iEEG (age $31.7 \pm 8.1$ years, 15 women, Table 1) performed the Posner peripheral cueing detection task[7] (Fig. 1A). Participants were asked to press a central button as soon as a target (an X) appeared within a left- or right-sided placeholder box. A non-predictive peripheral cue (a 100-ms thickening of the contour of one box) preceded the target with two possible stimulus onset asynchronies (SOA): 150 ms (short-SOA) or 600 ms (long-SOA) and appeared either on the same side of the target (Congruent trials) or opposite side (Incongruent trials) with equal probability.

Patients' performance was neurotypical[6,7], with a 30-ms IOR effect (Fig. 1B; 2-way-ANOVA: SOA X Congruence interaction, $F_{(1,27)} = 39.50$, $p < 0.001$, $\eta^2 = 0.164$; post-hoc test: long-SOA congruent vs. Incongruent $p < 0.001$). Congruent and incongruent RTs differed between SOAs (post-hoc tests: $p = 0.047$ and $p = 0.008$, respectively), but facilitation at short-SOA failed to reach significance ($p = 0.37$; see

**Table 1 | Implanted patient's demographic details**

| Patient # | Gender | Handedness | Number of electrodes (total 243) | Total number of contacts per patient (total 1884) | Implanted hemisphere |
|---|---|---|---|---|---|
| 1 | M | R | 10 | 104 | RH |
| 2 | F | R | 12 | 96 | LH + RH |
| 3 | M | R | 12 | 82 | RH |
| 4 | F | R | 10 | 82 | LH |
| 5 | M | R | 9 | 58 | RH |
| 6 | M | R | 11 | 90 | LH |
| 7 | F | R | 9 | 54 | LH |
| 8 | M | R | 9 | 63 | LH |
| 9 | M | L + R | 10 | 44 | LH + RH |
| 10 | M | R | 9 | 48 | LH |
| 11 | F | R | 10 | 88 | LH |
| 12 | F | R | 10 | 58 | RH |
| 13 | F | R | 8 | 76 | LH |
| 14 | F | R | 7 | 62 | LH |
| 15 | M | R | 10 | 70 | LH + RH |
| 16 | F | R | 9 | 78 | LH + RH |
| 17 | F | R | 8 | 61 | RH |
| 18 | M | R | 7 | 65 | RH |
| 19 | M | R | 7 | 31 | RH |
| 20 | M | R | 8 | 53 | LH |
| 21 | F | L | 8 | 56 | LH |
| 22 | M | L | 5 | 48 | LH |
| 23 | F | R | 8 | 63 | RH |
| 24 | F | R | 9 | 77 | RH |
| 25 | F | R | 9 | 67 | LH + RH |
| 26 | F | R | 9 | 54 | LH + RH |
| 27 | F | R | 12 | 93 | RH |
| 28 | M | R | 11 | 62 | LH |
| **Mean** | **54% F** | **89% R** | **9.1** | **67.3** | **57% RH** |

Fig. S1 for individual RT effects and target-side analysis), as is often the case with this subtle effect[8]. Moreover, left target Congruent RTs were slower than right target Congruent RTs, across both SOAs (Fig. S1B; repeated-measures 3-way ANOVA: Target-side X Congruence interaction- $F_{(1,27)} = 8.28$, $p = 0.008$, $\eta^2 = 0.007$), reflecting a Poffenberger effect[56,57], i.e., faster RTs for right-sided cue & target than for left-sided cue & target, when responding with the right hand. In Incongruent trials in which cue & target appear at opposite sides of the screen, this effect might have averaged out. No other target-side effects reached significance, and IOR and RT facilitation effects did not significantly differ between left-sided and right-sided targets (paired samples $t$-test; IOR side: $t(27) = 1.83$, $p = 0.077$; RT Facilitation side: $t(27) = 1.68$, $p = 0.11$). Catch trials were not statistically analyzed because of their small number, but patients never responded in those trials.

High-frequency broadband power (HFBB; 55–145 Hz) was extracted from 1403 usable contacts with bipolar montage, pooled across all participants (Fig. 2A; See Table 2 for detailed localization). Target-locked mean normalized HFBB activity was computed for each contact in the eight experimental conditions ($2 \times 2 \times 2$ design: SOA × Congruence × Ipsilateral/Contralateral target relative to contact; Fig. 2A).

The following steps were taken in the neural analysis approach. We first aimed to identify contacts with similar temporal activity across all conditions in a data-driven manner, using an adapted clustering trajectory k-means algorithm, which operated on the contact's target-locked temporal responses. We next explored the temporal progression of activity between the identified clusters. Given that the clusters were defined only based on their temporal dynamics, we then investigated the clusters' spatial localization, their white matter connectivity, and their spatial relations within the large-scale hierarchy of the cortical gradient, testing the prediction that meaningful clusters will group spatially in an ordered manner. We then turned to characterize how the neural activity across the clusters tracked visual, attentional, and response aspects of the Posner paradigm. Specifically, (1) we tested attentional effects by comparing neural activity across the attention contrasts used for the behavioral analysis; (2) we revealed response-related modulation by examining how differentiating target-locked activity according to the RT-affected neural activity; (3) we uncovered visual modulation of neural activity by applying the clustering anew to response-locked activity and studying how separating response-locked activity according to visual stimuli onset time influenced the clusters' neural activity. Finally, (4) we investigated whether the embedding of the cluster gradient in the cortical gradient extends beyond spatial topography and shares a functional hierarchy of temporal integration windows, which could correspond to a proposed theoretical mechanism underlying RT facilitation and IOR[6,41].

In order to reveal the main temporal patterns of activity that were sensitive to the experimental manipulations in a data-driven manner, we customized an unsupervised trajectory-clustering approach based on the k-means algorithm to cluster iEEG contacts according to their dynamic temporal patterns of activity across experimental conditions (Fig. S2). First, we selected responsive contacts, i.e., contacts with a significant effect in one condition or more, compared to baseline, which lasted at least 100 ms, for inclusion in the clustering analysis. This resulted in 644 responsive contacts, for each of which we calculated the temporal trajectory in the 8-dimensional condition space (Congruent/Incongruent Trial × short-SOA/long-SOA × Ipsilateral/contralateral target; see Fig. S2A, B), i.e., the path of each contact's HFBB over time across all experimental conditions. Each contact trajectory was then assigned to the cluster with the nearest trajectory-centroid by iteratively minimizing within-cluster Manhattan distances. For further analyses, we used a $k = 6$ solution, chosen using the Elbow method (see Fig. S2C, Fig. S3, and Table S1 for cluster number and stability across different k solutions, and Fig. S4A for the distribution of cluster contacts within participants).

Out of the chosen 6-cluster solution (Fig. 2A, B, Fig. S2C–E), we focused on three clusters of contacts that were stable across different k-solutions and whose activity patterns changed across the experimental conditions (Fig. 2B) and were positively correlated to one another, whereas their correlation with the other three clusters was negative or near zero, indicating that these clusters form a distinct group (Fig. S5).

The first cluster (Cluster 1; 68 contacts from 12 patients; Fig. 2B left, Fig. S4) showed early responses only to contralateral cues and targets. A second cluster (Cluster 2; 97 contacts from 18 patients; Fig. 2B middle) showed later ipsilateral and contralateral responses, with stronger responses to contralateral stimuli, demonstrating the spatial sensitivity of this cluster. The third cluster (Cluster 3; 67 contacts from 16 patients; Fig. 2B right) was the last to react, with stronger responses to bilateral targets than to cues, hence suggesting a sensitivity to task relevance. Importantly, the response in Clusters 2 and 3 was sensitive to the cue-target delay. For the short-SOA, cue and target responses were summed together, but they were segregated for the long-SOA. Activity in the three remaining clusters did not seem to vary across experimental conditions, with one cluster showing late inhibition, one showing late activation, and one showing no prototypical response (see Fig. S2D).

Next, we examined the temporal relationships between the clusters. The three target-locked clusters formed a temporal gradient

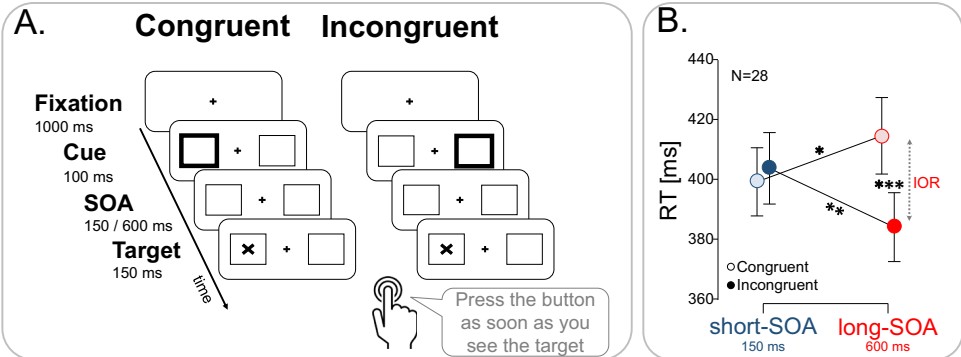

**Fig. 1 | Neurotypical performance of implanted patients in the Posner task.**
**A** Illustration of the Posner cued detection task. After 1000 ms of fixation, a cue (thickened placeholder) appeared for 100 ms at either side of the screen. On short SOA trials (short-SOA), the target (the letter X) occurred 150 ms after cue onset; on long SOA trials (long-SOA), the target appeared 600 ms after cue onset. The target appeared either on the same side of the screen as the cue (Congruent condition) or on the opposite site (Incongruent condition). Patients were required to press a central button with their right hand as soon as the target appeared while maintaining central fixation throughout the stimuli presentation. Catch trials ($n = 24$) had the same duration and cue presentation, but no target followed the cue. All trial types ($n = 336$) were equiprobable and randomly interleaved. Stimuli are not drawn to scale. **B** Patients' performance is neurotypical. Two-way-ANOVA, $*p = 0.047$; $**p = 0.008$; $***p < 0.001$. Error bars represent normalized SEM. $n = 28$ independent participants.

(Fig. 2C, D). The earliest activity emerged at Cluster 1, which peaked around $182 \pm 78$ ms post-target. Then followed Cluster 2 ($262 \pm 75$ ms post-target), and finally Cluster 3 ($383 \pm 141$ ms post-target; (Mixed 2-way ANOVA with Cluster and Congruence as factors; Cluster main effect $F(2,229) = 102.7$, $p < 0.001$, $\eta^2 = 0.378$; linear polynomial contrast: $p \leq 0.001$).

Having established a neural latency gradient between the three clusters, we then examined the spatial relationships between the clusters. Notably, the clustering was blind to the localization of the contacts. We thus hypothesized that meaningful clusters will tend to group anatomically. Cluster 1 mainly consisted of contacts in the bilateral occipitotemporal cortex and in the prefrontal cortex around the FEF (Fig. 3A top, Fig. S2G, and Supplementary Movie 1), consistent with its visual-like responses. Cluster 2 contacts were mainly in the caudal portion of the TPJ, around the angular gyrus, posterior temporal cortex, and prefrontal cortex (Fig. 3A middle, Fig. S2G and Supplementary Movie 1). The cluster was lateralized to the right hemisphere (See Supplementary Results and Fig. S2F, H). Cluster 3 was located mainly in the rostral TPJ region (around the supramarginal gyrus), posterior temporal cortex, and prefrontal cortex (Fig. 3A bottom, Fig. S2G and Supplementary Movie 1), and was lateralized to the left hemisphere (See Supplementary Results and Fig. S2F). Notably, the two latter clusters divided between them portions of known fronto-parietal attention networks[19,58].

We next asked if the contacts within each cluster were structurally connected. We divided each cluster's contacts into pre-rolandic contacts, located in the occipital, parietal and temporal lobes, and post-rolandic contacts, located in the frontal lobe, using the central sulcus as a landmark. A fiber tracking analysis paired with probability maps in 176 healthy individuals from the Human Connectome Project database[59] revealed that white matter tracts significantly connected pre-rolandic and post-rolandic contacts in the three clusters, suggesting these clusters' long-range contacts formed structural networks (Fig. 3D; threshold-free cluster enhancement-based non-parametric $t$-test, $p < 0.05$). We then examined the overlap of the connecting pre- and post-rolandic fibers with the three branches of the superior longitudinal fasciculus (SLF I; SLF II; SLF II), which connect the ventral and dorsal attention networks[19,35,60,61]. A probability cut-off of 50% was used for the SLF maps, and the resulting overlap was normalized to the number of cluster contacts per hemisphere. In Cluster 1, the connecting tracts mainly overlapped with SLF II in both hemispheres (Left hemisphere: SLF II 76.98%, SLF I 22.31%, SLF III 7.22%; Right hemisphere: SLF II 96.80%, SLF I 23.03%, SLF III 2.56%). In the right

hemisphere of the right-lateralized Cluster 2, there was a major overlap with SLF II, a smaller overlap with SLF III, and a minimal overlap with SLF I (SLF II 45.67%; SLF III 23.80%; SLF I 3.05%). An opposite pattern was found in the left hemisphere, where tracts overlapped with SLF III and had a smaller overlap with SLF II (SLF III 43.35%, SLF II 35.11%, SLF I 0.03%). In the left-lateralized Cluster 3, the connecting tract in the left hemisphere overlapped mainly with SLF III and had a small overlap with SLF II and a minimal overlap with SLF I (SLF III 36.78%; SLF II 28.45%; SLF I 0.65%). In the right hemisphere, Cluster 3 fibers were mainly associated with the SLF II and only minimally overlapped with SLF III and SLF I (SLF II 53.66%; SLF III 4.96%; SLF I 9.50%). These results suggest that the functional clusters identified solely based on their temporal responses correspond to well-defined structural networks.

We further asked if the clusters' anatomical localizations were ordered across large-scale cortical organization. We, therefore, explored how cluster localizations relate to the cortical gradient[51]. The position of a region along the gradient reflects its anatomical and functional cortical features[51,52] and can be described using a 2-dimensional coordinate system that represents the location along the early sensory and motor Periphery to the high-level multi-sensory Core[53]. Two main components define this 2-dimensional coordinate system: Dimension 1 extends from primary unimodal to transmodal regions, and Dimension 2 separates somatomotor and auditory cortices from the visual cortex[53]. Cluster 1 contacts were the most peripheral and closest to the visual end of Dimension 2; contacts in the Cluster 3 were the closest to the core, extending from the somatomotor end to transmodal regions (Dimension 1 contact values: 1-way ANOVA: $F(2,229) = 7.74$; $p < 0.001$, $\eta^2 = 0.06$; linear polynomial contrast: $p \leq 0.001$; Dimension 2 contact values: 1-way ANOVA: $F(2,229) = 77.79$; $p < 0.001$, $\eta^2 = 0.28$; linear polynomial contrast: $p \leq 0.001$; Fig. 3B, C). Thus, the clusters were embedded in the cortical gradient topography, forming a spatio-temporal gradient.

We then went on to study the way neural activity in the cluster gradient relates to attentional, visual and response aspects of the Posner task. We first explored how our experimental manipulation of attentional events influenced the clusters' target-locked neural activity. Specifically, we examined the neural correlates of the behaviorally significant IOR effect by comparing long-SOA Congruent and Incongruent trials in the cue time-window ($-600$ to $0$ ms) and in the target time-window ($0$–$800$ ms; time-resolved 3-way ANOVA with Congruence, Target Laterality and Contact Hemisphere as factors; Fig. 4, See Table S1 and Table S2 for full results).

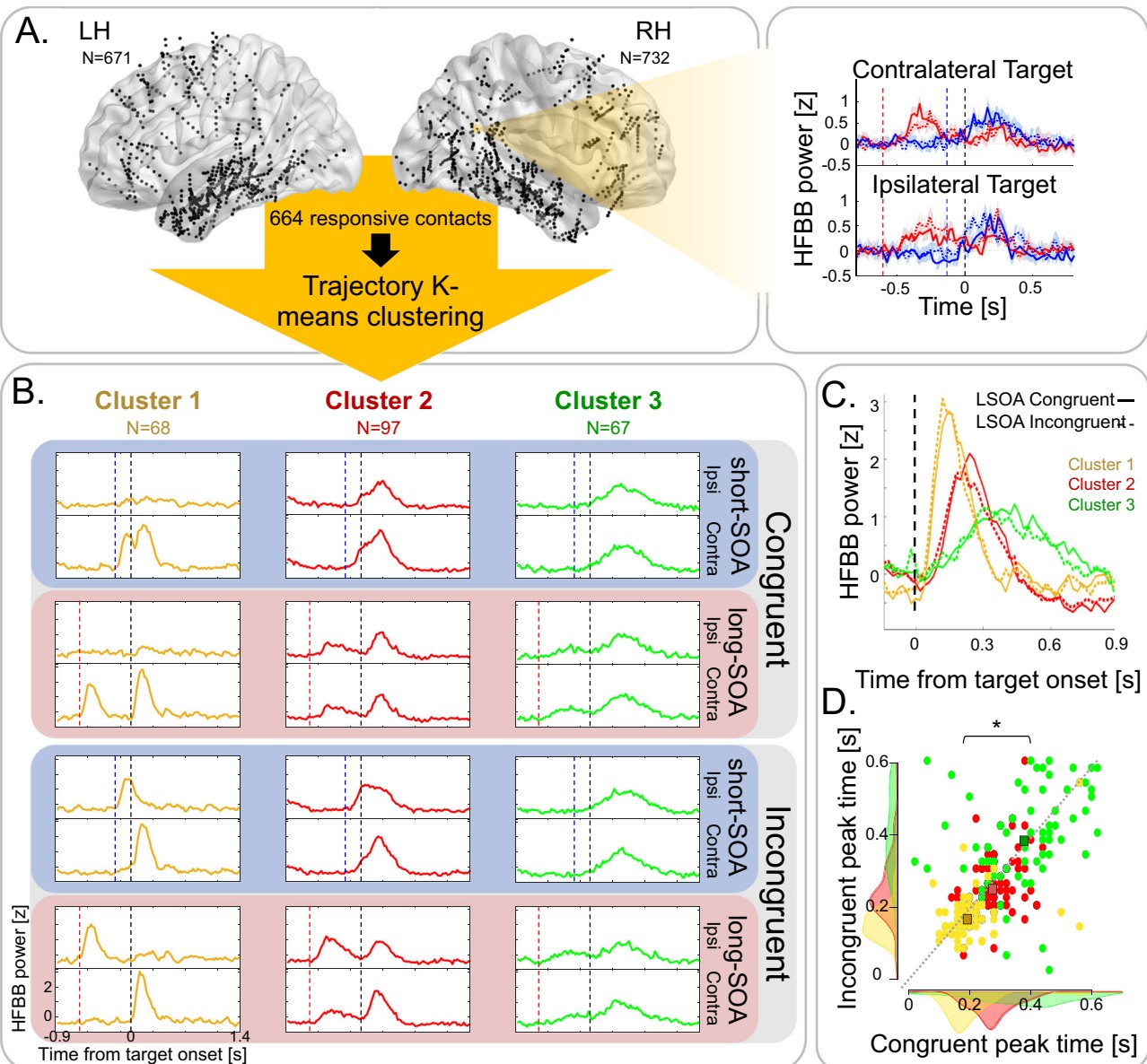

**Fig. 2 | Contact localization and trajectory clustering. A** Left: Illustration of the localization in normalized space (MNI152) of the contacts included in the analysis (black circles; *n* = 1403) in the left hemisphere (LH; *n* = 671) and in the right hemisphere (RH, *n* = 732), pooled across patients. Each localization is the mean coordinates of the two contacts composing the contact's bipolar montage. To reveal prototypical temporal patterns simultaneously across all conditions, the trajectories across the 8 condition dimensions of the mean high-frequency broadband (HFBB) target-locked activity of 664 significantly responsive contacts (significant time-point-by-time-point *t*-test for at least 100 ms in one of the experimental conditions compared to baseline), were clustered using a custom-made trajectory *K*-means approach. Right: Example of target-locked mean normalized HFBB responses of one contact in the right angular gyrus in Congruent (full lines) and Incongruent (dashed lines) trials, at short-SOA (blue) and long-SOA (red), with targets contralateral or ipsilateral to the contact. Dashed vertical lines represent onsets of the target (black), short-SOA (blue), and long-SOA (red) cues. Shaded areas represent SEM across trials. Brain visualization was done using BrainNet Viewer Matlab toolbox (Xia M, Wang J, He Y (2013) BrainNet Viewer: A Network Visualization Tool for Human Brain Connectomics. PLoS ONE 8(7): e68910. doi:10.1371/journal.pone.0068910). **B** Prototypical temporal profiles of contact clusters across conditions: Trimmed-mean target-locked activity profiles of three contact clusters across the 8 conditions (Congruent/Incongruent Trial × short-SOA/long-SOA × Ipsilateral target (Ipsi)/contralateral target (Contra)). Cluster 1 (yellow) shows contralateral fast responses, with cue-target activity segregation at both SOAs; Cluster 2 (red) shows bilateral slower responses with spatial sensitivity, with cue-target activity segregation at long-SOA but response integration in short-SOA; and Cluster 3 (green) shows bilateral slowest responses with stimulus-type sensitivity, with cue-target activity segregation at long-SOA but response integration at short-SOA. Dashed vertical lines represent target onset (black) and cue onset at short-SOA (blue) and long-SOA (red). **C** Temporal gradient of target-locked activity (trimmed-mean) of the three clusters. The Black dashed line depicts the target onset. **D** Scatter plot of peak times of mean target-locked activity of contacts of Cluster 1 (yellow circles), Cluster 2 (red circles), and Cluster 3 (green circles), in Congruent (*x*-axis) and Incongruent (*y*-axis) conditions, showing a significant temporal gradient (Mixed 2-way ANOVA, Cluster main effect *p* < 0.001, $\eta^2$ = 0.378; linear polynomial contrast: *p* ≤ 0.001). Squares represent mean peak time; the Dotted gray line denotes the equity line; Shaded areas represent peak time distributions.

**Table 2 | Responsive electrode localization according to the Desikan–Killiany–Tourville atlas[91]**

| Region name | Responsive electrodes N | Cluster 1 N | Cluster 2 N | Cluster 3 N |
|---|---|---|---|---|
| Banks superior temporal sulcus | 9 | 1 | 4 | 1 |
| Caudal anterior-cingulate cortex | 3 | 0 | 0 | 0 |
| Caudal middle frontal gyrus | 12 | 2 | 2 | 1 |
| Entorhinal cortex | 6 | 0 | 0 | 0 |
| Fusiform gyrus Posterior | 33 | 7 | 8 | 3 |
| Fusiform gyrus Med | 14 | 2 | 2 | 0 |
| Fusiform gyrus Anterior | 10 | 0 | 0 | 0 |
| Inferior parietal cortex | 51 | 19 | 14 | 5 |
| Inferior temporal gyrus Posterior | 28 | 1 | 8 | 1 |
| Inferior temporal gyrus Middle | 14 | 0 | 3 | 0 |
| Inferior temporal gyrus Antrior | 13 | 0 | 0 | 0 |
| Lateral occipital cortex | 20 | 6 | 5 | 2 |
| Lingual gyrus | 17 | 1 | 0 | 3 |
| Medial orbital frontal cortex | 4 | 0 | 0 | 0 |
| Middle temporal gyrus Posterior | 37 | 10 | 12 | 1 |
| Middle temporal gyrus Middle | 19 | 0 | 2 | 0 |
| Middle temporal gyrus Anterior | 35 | 0 | 0 | 0 |
| Parahippocampal gyrus | 8 | 0 | 0 | 0 |
| Paracentral lobule | 1 | 0 | 0 | 0 |
| Pars opercularis | 8 | 0 | 0 | 1 |
| Pars orbitalis | 36 | 0 | 0 | 0 |
| Pars triangularis | 9 | 0 | 0 | 4 |
| Pericalcarine cortex | 1 | 0 | 0 | 0 |
| Postcentral gyrus dorsal | 1 | 0 | 0 | 0 |
| Postcentral gyrus ventral | 1 | 0 | 0 | 0 |
| Posterior-cingulate cortex | 3 | 0 | 1 | 1 |
| Precentral gyrus dorsal | 16 | 6 | 3 | 4 |
| Precentral gyrus ventral | 5 | 0 | 3 | 1 |
| Precuneus cortex | 1 | 0 | 0 | 0 |
| Rostral middle frontal gyrus | 16 | 0 | 4 | 2 |
| Superior frontal gyrus | 46 | 0 | 8 | 16 |
| Superior parietal cortex | 10 | 1 | 3 | 1 |
| Superior temporal gyrus Posterior | 19 | 2 | 1 | 3 |
| Superior temporal gyrus Middle | 17 | 0 | 0 | 0 |
| Superior temporal gyrus Anterior | 13 | 0 | 0 | 3 |
| Supramarginal gyrus | 22 | 0 | 3 | 9 |
| Temporal pole | 14 | 0 | 0 | 0 |
| White matter | 49 | 10 | 10 | 5 |
| hippocampus | 18 | 0 | 1 | 0 |
| amygdala | 5 | 0 | 0 | 0 |

In the cue time-window, Congruent and Incongruent trials did not significantly differ overall (no significant main Congruence effect; Fig. S6), reflecting the fact that the cue location did not predict the congruence of the upcoming target. Instead, there were mainly neural effects reflecting the differential lateralization of cues preceding Congruent and Incongruent targets (See Supplementary Material).

In the target time-window, cluster 2 showed a Congruence main effect at the offset of the target-related activity (240–300 ms post target; largest $p = 0.002$; see Fig. 4D for examples of single contacts). Moreover, in the contacts of this cluster in the right hemisphere, the response peaked 22 ms later in the Congruent than in the Incongruent trials (140–220 ms post target onset; Hemisphere x Congruence interaction: largest $p = 0.03$; post hoc tests: largest $p = 0.014$), mirroring behavioral IOR. There were no congruence effects in Cluster 1 (Fig. 4A), and in Cluster 3, there was only a late Congruence effect at 660–680 ms post target (largest $p = 0.003$). Therefore, IOR-related activity was mainly restricted to Cluster 2, thus attentional events corresponded to the neural dynamics of this cluster.

Despite the lack of a significant behavioral effect of RT facilitation, the effect might be masked by other processes, as is often the case[6,8,11]. Current theories postulate that even when masked, the facilitation effect nevertheless exists[6,8,11]. We therefore performed an exploratory time-resolved ANOVA analysis with the factors Congruence, Target-side, and Hemisphere to test the attentional neural effect in Cluster 2 also in the short-SOA. In the target time-window, Cluster 2 showed a significant Congruence × Target-side interaction effect (−60 to 140 ms post target; largest $p = 0.022$) and a main Target-side effect (160–300 ms; 320–360 ms; 440–460 ms post target onset; largest $p = 0.012$; see Fig. S7). This reflects a combination of stronger responses for contralateral stimuli (cue or target), which are summed together, leading to a faster and stronger activation for contralateral congruent compared to contralateral incongruent cues and targets and compared to ipsilateral ones. This differential summed activity translates to a neural preference for stimuli repeating in the same specific spatial (contralateral) location, dovetailing the behavioral RT facilitation effect, in which RT is faster for repeated stimuli in a specific location.

The observed differences between SOA and Congruence conditions across clusters could be explained by different theta phases at target onset[62,63], as the neural activity at the short-SOA and long-SOA could fall into opposite phase bins. A control mixed ANOVA analysis revealed that theta phase could not explain these effects, either across the entire sample of contacts or when looking at particular clusters of contacts (see Supplementary Results). A Bayesian ANOVA confirmed these negative findings, which are consistent with a recent paper that found no evidence for rhythmic sampling in inhibition of return behavioral effects[64].

How do these clusters of neural activity relate to the manual response? We examined whether cluster neural dynamics relate to motor response timing across experimental conditions, reflecting the significant RT differences between SOAs in the Congruent and Incongruent conditions. In each cluster, we divided the trials (pooled across conditions) into 20 quantiles according to their RT (Fig. 5A) and tested the relation of RT-bins with the neural activity using a time-resolved 1-way repeated measures ANOVA (See Fig. 5A, B for results and examples of single contacts). In Cluster 2, the offset of the target-related activity differed across RT bins (300–560 ms post target; largest $p = 0.028$), with a faster decay at faster RT bins, just before the motor response. In Cluster 3, an RT-bin effect occurred around the peak of target-related activity and button-press time (280–300 and 400–420 ms post target; largest $p = 0.007$). In Cluster 1, an RT-bin effect occurred at 500–540 and 560–680 ms post target onset ($p < 0.002$), suggesting an RT-related late modulation after response offset and button press time. RT-related target-locked activity in Clusters 2 and 3 was confirmed by cross-correlation analysis (See Supplementary Results and Fig. S9),

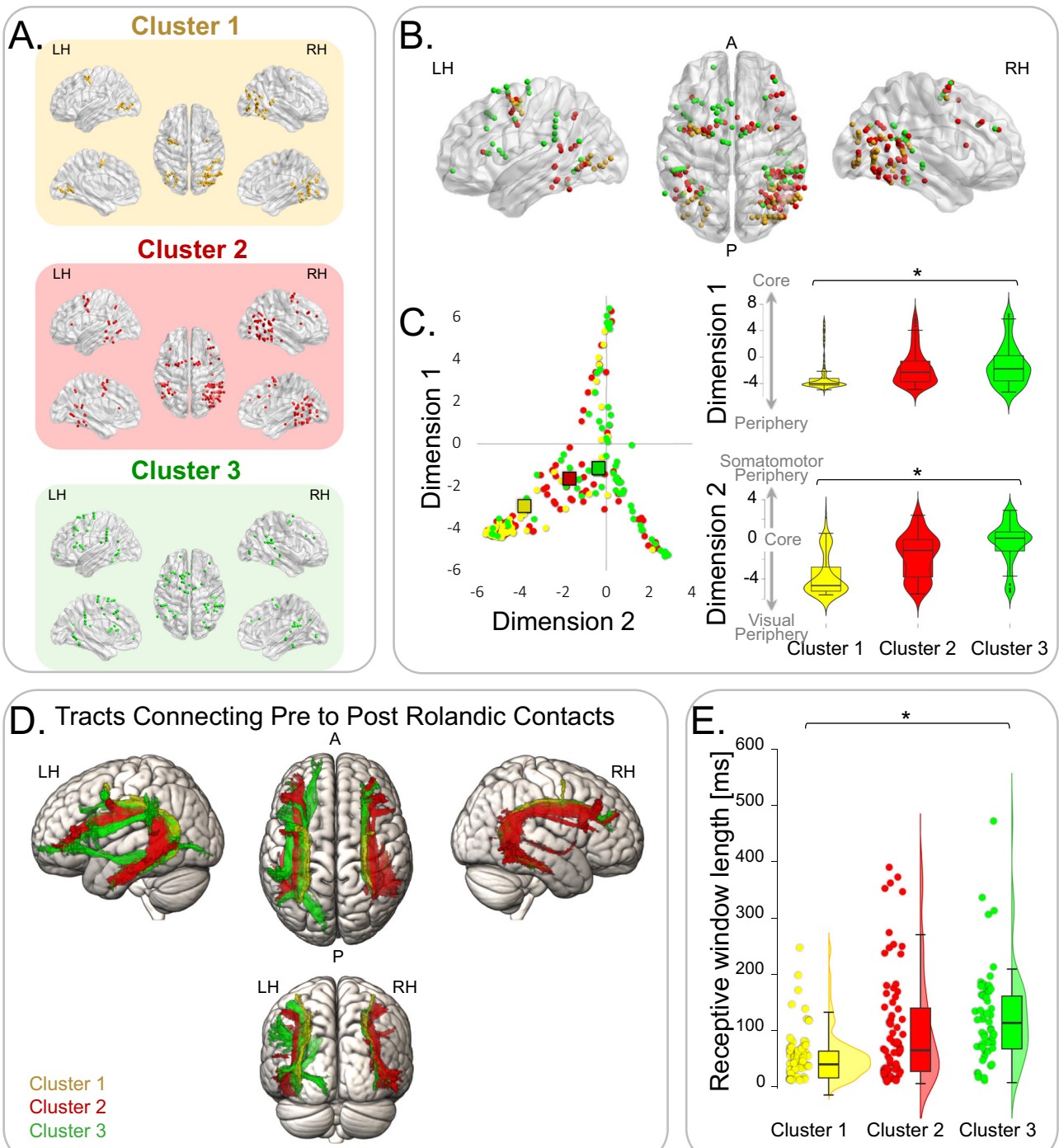

which revealed that only in these clusters did the temporal dynamics of neural activity shift according to RTs and that this shift correlated with RTs. Thus, neural activity in Clusters 2 and 3 was related to the timing of the upcoming motor response, reflecting the behavioral outcome of the task and its associated neural processes.

We next studied the neural correlates of the visual aspects of the Posner task by adopting a complementary approach and examining the visual modulation of response-locked activity. To avoid biases, we applied the trajectory k-means clustering analysis to response-locked activity (Fig. S10A–C and Supplementary Movie 2) instead of using the clusters obtained based on the target-locked activity. To map the correspondence of the seven response-locked clusters to the previously identified target-locked clusters, we performed a contingency analysis that revealed four corresponding response-locked clusters

$(\chi^2_{(30)} = 1442; p < 0.001;$ Contingency coefficient 0.83; Fig. 5 and S10D). Specifically, locking the activity to the response further separated the clusters: RT-Cluster 1 (46 contacts; 60.3% of target-locked Cluster 1), RT-Cluster 2a (85 contacts; 35.3% of target-locked Cluster 1 and 49.5% Cluster 2), RT-Cluster 2b (79 contacts; 46.4% of target-locked cluster 1 and 31.3% of Cluster 2), and RT-Cluster 3 (39 contacts; 50.7% of target-locked Cluster 3). We repeated the RT-binning analysis, as described above (Fig. S8B), and tested the RT-bin effect on the neural activity using a time-resolved 1-way repeated measures ANOVA (See Fig. 5C, D for results and examples of individual contacts). The response-locked clusters showed a spatiotemporal gradient and mapped onto the cortical gradient topography, similar to the target-locked clusters. (Fig. S11). Notably, locking activity to the response allowed separation of the peripheral RT-Cluster 2a contacts from the RT-Cluster 2b

**Fig. 3 | Clusters exhibit a spatiotemporal gradient. A** Clusters' spatial profile. Illustration of the localization of the contacts composing each cluster: Cluster 1 (yellow), Cluster 2 (red), Cluster 3 (green). For each cluster, dots represent contacts' localization in dorsal (middle), lateral (top), and medial (bottom) views of the right hemisphere (RH; right) and of the left hemisphere (LH; left). **B** Core–Periphery gradient: Clusters' anatomical localization follows core–periphery gradients[53], where Cluster 1's contacts are the most peripheral, and Cluster 3's contacts are closest to core regions. **C** Left: Scatter plot of contacts localization along core–periphery gradients (Cluster 1–yellow circles, $n = 62$ independent contacts; Cluster 2–red circles, $n = 97$ independent contacts; Cluster 3–green circles, $n = 67$ independent contacts; rectangles represent clusters' mean). Right: Violin plots of contacts localization along Core-Periphery gradients for Cluster 1 (yellow), Cluster 2 (red) and Cluster 3 (green), showing a significant core-periphery gradient (Gradient 1: 1-way ANOVA, $p < 0.001$, $\eta^2 = 0.06$; linear polynomial contrast: $p \le 0.001$; Gradient 2: 1-way ANOVA, $p < 0.001$, $\eta^2 = 0.28$; linear polynomial contrast: $p \le 0.001$; $n = 232$ independent contacts in total). The box centerlines depict the medians, the

bounds of the box depict the 75%/25% quartiles, and the whiskers depict the top & bottom 25% percentiles. **D** Cluster contacts are structurally connected: Corrected tractography t-maps, showing the significant white matter voxels, which connect pre and post-rolandic contacts within each cluster (Cluster 1–yellow; Cluster 2– red, Cluster 3–green), derived from a fiber tracking analysis of 176 healthy individuals. **E** Contacts' receptive windows lengthen along the cluster gradient: Raincloud plots of individual contacts' receptive window length (circles), showing a significant linear lengthening from Cluster 1 (yellow, $n = 62$ independent contacts), to Cluster 2 (red, $n = 97$ independent contacts), to Cluster 3 (green, $n = 67$ independent contacts; 1-way ANOVA: $p < 0.001$, $\eta^2 = 0.11$; linear polynomial contrast: $p \le 0.001$; $n = 232$ independent contacts in total). The box centerlines depict the medians, the bounds of the box depict the 75%/25% quartiles, and the whiskers depict the top & bottom 25% percentiles. Brain visualization was done using BrainNet Viewer Matlab toolbox (Xia M, Wang J, He Y (2013) BrainNet Viewer: A Network Visualization Tool for Human Brain Connectomics. PLoS ONE 8(7): e68910. doi:10.1371/journal.pone.0068910).

contacts, which were closer to the core (Fig. S11D). Because RT is defined as the time from target onset to the response, this procedure sorted the response-locked trials according to target onset and thus could unveil visual modulation of response-locked activity. The onset of the response-locked activity was modulated by target onset only in RT-Cluster 1 (120–100 ms pre-response; largest $p = 0.04$) and RT-Cluster 2a (700–680 ms, 520–500 ms, 300–200 ms pre-response; largest $p = 0.004$). In RT-Cluster 2b and RT-Cluster 3, the neural activity peak was aligned to the response without significant visual modulation. The visual modulation of response-locked activity in RT-Cluster 1 and RT-Cluster 2a was confirmed by cross-correlation analysis (See Supplementary Results and Fig. S12), which revealed that only for contralateral targets in these clusters the temporal dynamics of neural activity was shifted according to target-onset and this shift correlated with target-onset time. Thus, response-locked activity revealed that only the clusters with early response-locked activity showed visual modulation, while clusters with later activity were only sensitive to the timing of the motor response.

Finally, we investigated whether the embedding of the cluster gradient in the cortical gradient extends beyond spatial topography and shares a functional hierarchy with it. Importantly, one of the features that change along the cortical gradient is the length of temporal receptive windows (TRW, i.e., the time window in which previously presented information can affect the processing of a newly arriving stimulus), which lengthen and integrate over longer durations when moving up the gradient[51,52,65,66]. Temporal integration was suggested as a potential mechanistic computation underlying RT facilitation and IOR[6,41]. Therefore, we asked if TRWs also lengthen along the cluster gradient. We estimated TRW length by calculating the decay time constant of the autocorrelation function applied to the non-filtered neural time series for each contact in the three clusters[66,67]. TRW length increased when moving up the cluster gradient (Fig. 3E; TRW length: Cluster 1 to $54.33 \pm 44.96$; Cluster 2 to $102.56 \pm 99.15$; Cluster 3 to $124.91 \pm 87.13$; 1-way ANOVA: $F(2,103.98) = 17.83$; $p < 0.001$, $\eta^2 = 0.113$; linear polynomial contrast: $p \le 0.001$), suggesting that along this trajectory, integration is over longer durations[51,68,69]. Hence, the cluster gradient shares a similar temporal integration hierarchy with the cortical gradient[51,66], mirroring the pattern of integration/segregation of cue-target neural responses observed along the cluster gradient.

## Discussion

Here we aimed to establish how attention-capturing events modulate visual, attentional, and response-associated neural processing in the human brain and how the involved brain networks map onto the large-scale cortical topography. Overall, we provide a high-resolution, comprehensive depiction of the cortical dynamics underlying human exogenous attention. Our findings reveal that attentional events

differentially define neural activity along a series of clusters, which form a spatiotemporal gradient, extending from the visual cortex to frontoparietal regions. This gradient is embedded in the periphery-core cortical topography, which is a primary organizing axis of the human cerebral cortex[51,53,55]. Cluster neural activity at one end of the gradient is modulated by visual attributes, while activity at the gradient's other end reflects the timing of the upcoming response, with attentional modulations occurring at the intersection of visual and response signals. Notably, temporally close stimuli elicit discrete neural responses at the visual end of the gradient, yet at its frontoparietal end, they elicit a single pooled neural response. Moreover, TRWs lengthen along the cluster gradient, like the hierarchy of timescales along the cortical topography in which the clusters are embedded. These findings stress the importance of studying fast and dynamic cognitive processes with high-resolution methods and suggest that attention is not a discrete multi-step operation but rather arises over large neural gradients embedded in the cortical topography, along which perceptual and response-related signals integrate.

We identified three key components along exogenous attention's cortical gradient. The first, Cluster 1, is situated at the peripheral end of the cortical gradient, encompassing the occipitotemporal cortex[70], and the vicinity of the FEFs[71], where ultra-fast visual activation was reported[72]. Its occipital and FEF-adjacent contacts were structurally connected mainly by the middle branch of the SLF (SLF II). Functionally, it only responded to contralateral visual stimuli, and its neural responses to the cue and target were segregated, even at the short cue-target delay.

Clusters 2 and 3 are located closer to core regions of the cortical gradient and overlap with known frontoparietal attention networks[19,58]. The neural activity in Cluster 2, occurring midway along the gradient, is sensitive to cue-target spatial positions and delays and exhibits IOR-related onset and offset. Both visual processing of the target and manual response preparation shape the neural activity in this cluster, which is lateralized to the right hemisphere, consistent with lesion and neurostimulation data on IOR[28–30,33,34]. Despite the fact that we did not find a significant behavioral effect of RT facilitation, the involvement of Cluster 2 neural activity in attentional computation in the short-SOA condition is plausible. First, RT facilitation is an elusive effect, easily masked by other processes[6,8,11]. Our design was not optimal for unmasking the behavioral effect because of the lack of temporal overlap between cue and target, which is one of the conditions that favor the appearance of RT facilitation in detection tasks[8,19]. In addition, the Poffenberger effect we observed further masked the RT facilitation effect. Yet, current theories postulate that facilitation exists[6] even when it is behaviorally offset by IOR, which is always present with peripheral cues, even at short SOAs. Our exploratory analysis revealed in Cluster 2 at the short-SOA a differential cue-target summed activity, which translates to a neural

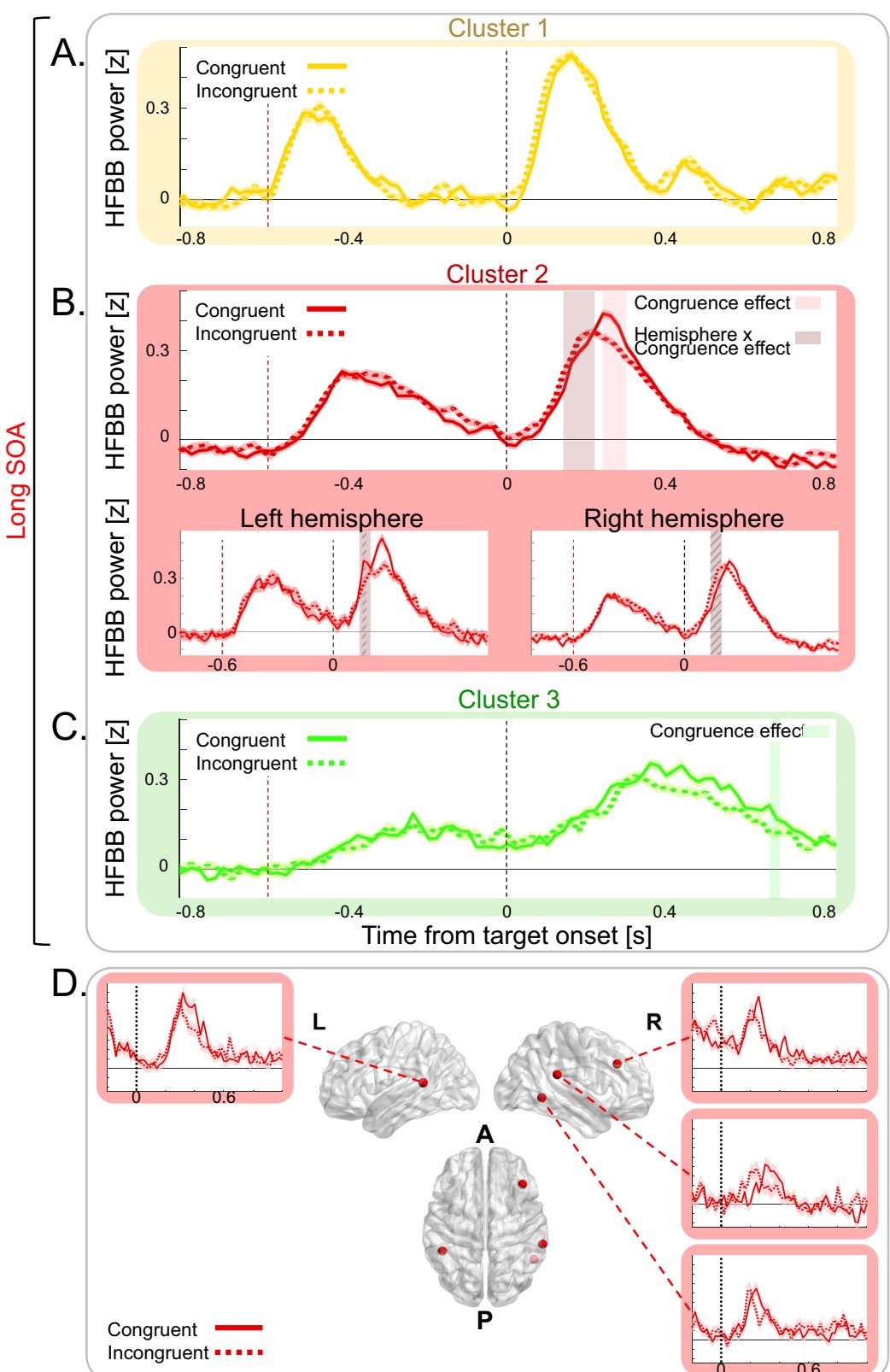

preference for stimuli repeating in the same specific spatial contralateral location. This neural effect dovetails with the behavioral RT facilitation effect, in which RT is faster for repeated stimuli in a specific location. Therefore, our results suggest that the activity in Cluster 2 represents a key attentional processing of exogenous cueing effects in both short and long SOAs, associating perception and action signals.

On the other hand, neural activity in Cluster 3 shows sensitivity to stimulus identity, with stronger activation for response-requiring targets than for cues. It is lateralized to the left hemisphere, contralateral to the responding hand, and its response-locked activity peaks at the time of the motor response, which also modulates its target-locked activity. Furthermore, this cluster is anatomically situated between the somatomotor end and transmodal core regions of the core-periphery

**Fig. 4 | IOR-related neural activity.** Mean target-locked long-SOA activity in Cluster 1 (yellow), Cluster 2 (red), and Cluster 3 (green), was computed over trials pooled across all cluster contacts for Congruent trials (full lines) and Incongruent trials (dashed lines). **A** In Cluster 1, no significant Congruence effect was observed in a 3-way ANOVA with Holm multiple comparisons correction. **B** In Cluster 2 activity in Congruent and Incongruent trials (IOR-related) differed significantly in a 3-way ANOVA with Holm multiple comparisons correction at 0.24–0.3 s post target (shaded red areas; Congruence main effect: largest $p = 0.002$), and a significant hemispheric difference between IOR-related responses was observed at 0.14–0.022 s post target (shaded brown area; Hemisphere × Congruence interaction: largest $p = 0.03$; Diagonally striped areas represent significant Congruence × Hemisphere post hoc comparisons ($p < 0.05$)). **C** In Cluster 3, activity in Congruent and Incongruent trials differed significantly in a 3-way ANOVA with Holm multiple comparisons correction at 0.66–0.68 s post target (green shaded area; Congruence main effect: largest $p = 0.003$). A-C. Shaded areas around traces depict SEM; Dashed vertical lines represent target onset (black) and cue onset (red) at the long-SOA Condition. **D** Representative examples of HFBB power IOR-related activity in the Congruent (full line) & Incongruent (dashed line) long-SOA conditions of individual contacts of Cluster 2, shaded areas around traces depict SEM. $p$ Values are Holm corrected. Brain visualization was done using BrainNet Viewer Matlab toolbox (Xia M, Wang J, He Y (2013) BrainNet Viewer: A Network Visualization Tool for Human Brain Connectomics. PLoS ONE 8(7): e68910. doi:10.1371/journal.pone.0068910).

gradients. Because the patients only responded with their right hand, we cannot completely rule out that the left hemisphere response is simply stronger, and thus the cluster's activity is not related to response aspects of the task. However, this cluster contains right hemisphere contacts as well, and its contacts are also localized in non-motor regions, such as the posterior temporal lobe and supramarginal gyrus. This fact, together with the entire line of evidence mentioned above, supports the suggestion that Cluster 3 encodes decisional and response aspects.

Along all this gradient of clusters, neural activity shows spatial sensitivity, sensitivity to cue-target delay, sensitivity to task relevance, and association with RT, therefore encoding the information necessary to underlie exogenous attention RT effects such as IOR, which depend on the delay and co-localization of attentional events.

Importantly, these findings depart from traditional attention models of multi-step processing across visual areas. Instead, exogenous attentional effects seem to emerge along a continuous neural trajectory of large-scale cortical gradient, which bridges perceptual and response processing. These findings reconcile long-debated theories about the perceptual-motor (or input-output) dichotomy of attentional processes[10,11,73]. We find both perceptual and motor effects; however, they form a gradient rather than a dichotomy. These findings dovetail with the idea that attention organizes the activity of sensory and motor networks, generating alternating states for sampling sensory information versus shifting attention and responding[63].

Despite the overlap of Clusters 2 and 3 with known frontoparietal attention networks, their anatomy and function diverge from neurophysiological models of human attention (e.g.,[19]). First, in the TPJ, which constitutes a single node of the right-lateralized ventral attention network[19], these clusters occupy distinct portions, which differ in their functional and structural connectivity[42,61,74,75]. The caudal TPJ portion (Cluster 2) connects to the superior frontal gyrus/FEF of the dorsal attention network[42,61,75] through the middle branch of the SLF (SLF II) and thus provides direct communication between the ventral and dorsal attention networks.

In contrast, the rostral TPJ (Cluster 3) is connected to the middle and inferior frontal gyri through the ventral branch of the SLF (SLF III), thus linking nodes of the ventral attention network. Both SLF II and SLF III show anatomical or functional lateralization to the right hemisphere[61], and their inactivation or disconnection was associated with signs of left spatial neglect[33,35]. Indeed, our findings demonstrate that temporoparietal and prefrontal contacts in Clusters 2 and 3 are connected by the SLF, and our overlap analysis suggests that in the right hemisphere, the right-lateralized Cluster 2 is more connected by the SLF II, while the left-lateralized Cluster 3 is more connected by the SLF III in the left hemisphere. Yet because of the overlap between probabilistic maps of SLF II and III templates, these latter findings should be validated in future studies exploring neural activity and tractography in the same sample of participants.

Similarly, Clusters 1, 2, and 3 encompass contacts in the dorsolateral prefrontal cortex, indicating that when examining in sufficient spatio-temporal resolution, this region, which constitutes a single node of the dorsal attention network[19], can be dissociated into distinct networks.

Furthermore, our findings localizing contacts from Cluster 2 and 3 to the posterior temporal lobe, a region outside the scope of hallmark attention models[19,58], suggest that this area may contribute to exogenous attention processing, dovetailing recent studies in humans and non-human primates[76,77].

Functionally, our findings suggest that contrary to these models, not only do the prefrontal nodes of the dorsal attention network process information pertaining to the contralateral visual field[42,78] but rather respond to stimuli in both contralateral and ipsilateral visual fields. Conversely, the activity recorded in contacts in the TPJ belonging to Cluster 2 presented spatial sensitivity, contrary to the assumption of some models that this functional region lacks spatial mapping[19]. Additionally, our findings concerning the TPJ are not completely consistent with the prominent Corbetta and Shulman model[19]. Based on fMRI data, this model postulates that exogenous orienting does not activate the TPJ, which only responds to reorienting to response-relevant targets. Corbetta and Shulman[19] suggest that when an important stimulus appears outside the current focus of attention, fast-latency signals from the ventral network initiate reorienting by sending a "circuit-breaking" or interruption signal to dorsal regions, which changes the locus of attention. In other words, according to this model, TPJ should not respond to peripheral non-informative cues, only to unexpected incongruent targets. However, we found that TPJ contacts were also activated in response to cues and also incongruent trials when the target location corresponded to the location of the preceding cue, aligning with previous causal evidence from TMS studies[79,80]. Therefore, our findings suggest that the TPJ is not just a circuit breaker responding when unexpected and pertinent targets appear and reorienting of attention is needed[19].

What are the cortical characteristics that favor the localization of attentional processing to a particular extent of the cluster gradient? Besides the convergence of perceptual and response signals, a potential factor may be the temporal integration properties of the involved regions. This trait changes in a continuous manner along the temporal hierarchy of TRWs, a key feature of the core-periphery gradient, analogous to the spatial hierarchy of receptive fields[51,52,65,68,69,81,82]. Thus, along this gradient, integration is over longer durations, and selectivity for coherent temporal structures increases[51,65,68,69]. TRW length is intrinsically determined by a region's cytoarchitecture and macro- and micro-circuit connectivity[52,69]. Such a hierarchy of TRWs could enable a dynamic interaction with a continuously changing environment, with fast fluctuations associated with sensory processing at the bottom of the hierarchy, and slow fluctuations, which reflect contextual changes in the environment, at the hierarchy top[69]. Moreover, a hierarchy of TRWs can serve as a scaffold for putative recurrent temporal computations that support neuronal sensitivity to sequential events and boost robustness to changes in input gain and timing, such as temporal pooling, i.e., the integration of prior information across the TRW[68]. Indeed, recent evidence showed that TRWs could serve cognitive functions[52,83,84]. For example, prefrontal cortex TRWs expanded during working memory maintenance and predicted individual performance[52]. Correspondingly, our finding that TRWs lengthen along the cluster gradient reveals potential

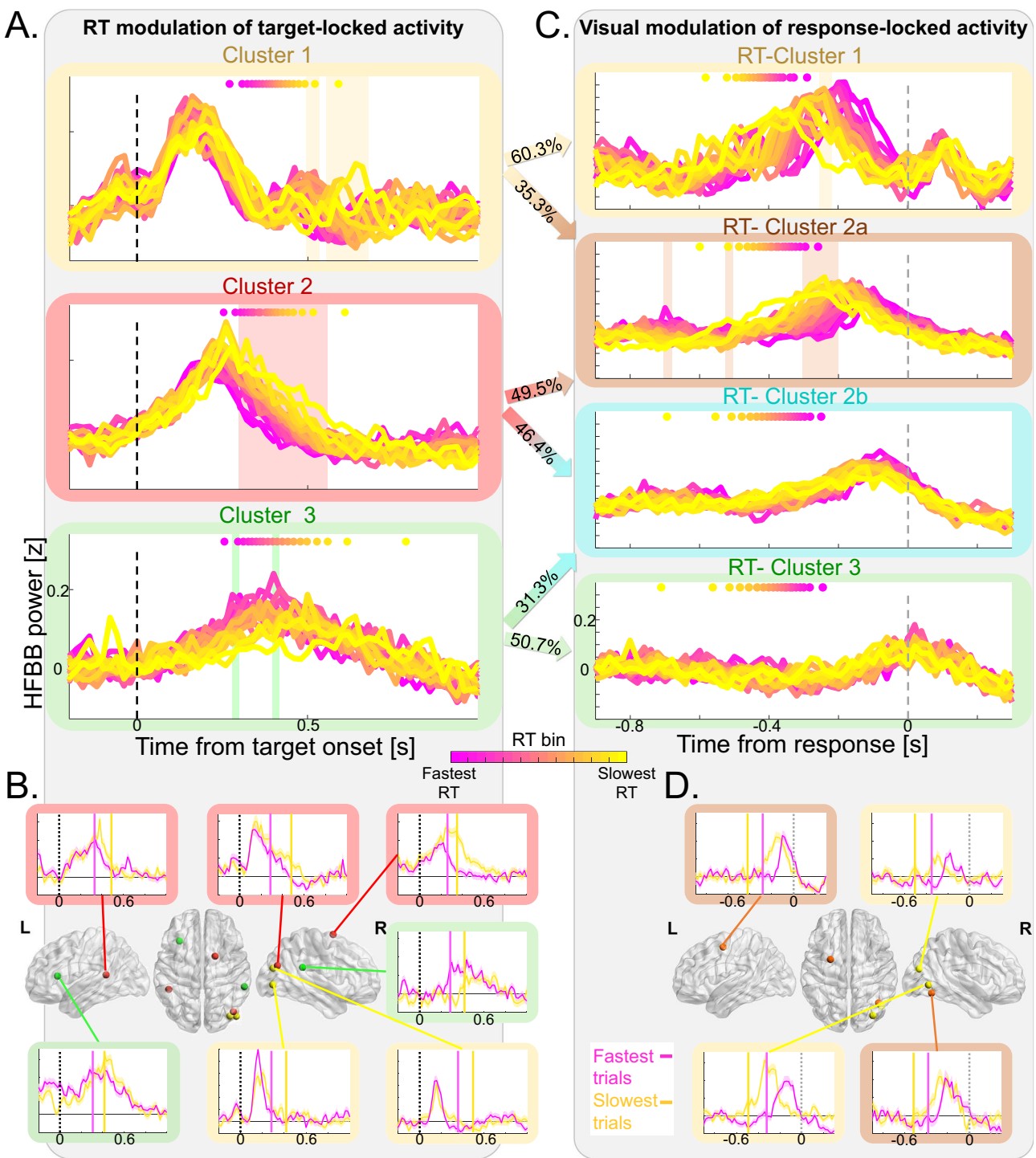

temporal operations on the basis of exogenous attention. Further-more, the integration of cue-target responses in Clusters 2 and 3 in the long-SOA could reflect temporal pooling[68]. In Cluster 1, situated lower on the gradient, TRWs are shorter, allowing for segregation of activity even at short delays. In upstream frontoparietal clusters where TRWs are longer, cue- and target-induced responses resulted in a single activity peak. This temporal pooling might group the cue and target in a single event[47], leading to RT facilitation at short cue-target delays[6,41,85]. These findings dovetail with the hypothesis that RT facilitation results from a summation of cue-related and target-related responses, thus reflecting hard-wired limitations of the neural system that cannot respond separately to rapidly repeated stimuli and processes them as a single event[6,41,85]. According to cue-target event

integration-segregation hypothesis[6], RT facilitation arises when the net effect of facilitatory processes, such as exogenous spatial attention orienting and binding-associated spatial selection benefit, is larger than the detection cost the binding might cause due to the difficulty in detecting the onset of the second bound stimulus[6]. Longer cue-target delays could instead provide the system with enough time to segregate cue- and target-related responses[6,41]. Hence, our results contribute to resolving the longstanding debate surrounding the nature of IOR. In Clusters 2 and 3, IOR was linked to the segregation of neural responses, with distinct peaks corresponding to cues and targets. Notably, in Cluster 2 (encompassing the angular gyrus and lateral prefrontal cortex), the timing of these distinct peaks, as well as their decay, mirrored behavioral IOR. Consequently, our findings provide a refined

**Fig. 5 | RT & visual modulation of target-locked & response-locked neural activity. A** RT modulates target-locked neural activity, pooled across conditions and color-coded from fastest (Magenta) to slowest (yellow) RT bin. A dashed vertical black line represents target onset; Color-coded dots at the top of each panel represent mean RT for each bin (pink−fastest RT to yellow−slowest RT); 1-way repeated measures ANOVA, Holm multiple comparisons correction. Top: Late RT modulation of activity in Cluster 1 (yellow): Main effect of RT bin at 0.5–0.54 and 0.56–0.68 s post-target (shaded yellow area; largest $p = 0.002$). Middle: RT modulation of neural response offset in Cluster 2 (red): Main effect of RT bin at 0.3–0.56 s post target (shaded red area; largest $p = 0.028$). Bottom: RT modulation of response in Cluster 3 (green): Main effect of RT bin at 0.28–0.3 and 0.4–0.42 s post target (shaded green area; largest $p = 0.007$). **B** Examples of single contact neural activity in the fastest (pink) and slowest (yellow) thirds of trials for the three target-locked clusters. Vertical dashed black lines represent target onset; Vertical full lines denote mean RT for fastest (magenta) and slowest (yellow) trials, shaded areas around traces depict SEM. **C** Visual modulation of response-locked neural activity pooled across conditions, color-coded from fastest (Magenta) to the slowest (yellow) bin. The dashed vertical gray line represents RT; color-coded dots

at the top of each panel represent the mean target onset time for each bin (pink−earliest onset to yellow−latest onset); 1-way repeated measures ANOVA, Holm multiple comparisons correction. Top: target onset time modulates activity in the RT-Cluster 1 (yellow): Main effect of RT-bin at 0.12–0.10 s pre-response (shaded yellow area; largest $p = 0.04$). Target onset time modulates activity in the RT-Cluster 2a (orange): Main effect of RT bin at 0.70–0.68 s, 0.52–0.50 s, and 0.30–0.20 s pre-response (shaded orange area; largest $p = 0.004$). No significant modulation in RT-Cluster 2b (turquoise) and RT-Cluster 3 (green). Arrows between panels **A** and **C** denote the contingency between target-locked and response-locked clusters (see Fig. S10). **D** Examples of single contact neural activity in the fastest (pink) and slowest (yellow) thirds of trials for RT-Cluster 1 and RT-Cluster 2a. Vertical dashed gray lines represent RT; Vertical full lines denote the mean target onset time for the fastest (magenta) and slowest (yellow) trials, shaded areas around traces depict SEM. $p$ Values are Holm corrected. Brain visualization was done using BrainNet Viewer Matlab toolbox (Xia M, Wang J, He Y (2013) BrainNet Viewer: A Network Visualization Tool for Human Brain Connectomics. PLoS ONE 8(7): e68910. doi:10.1371/journal.pone.0068910).

anatomical and functional specification of earlier results obtained from studies involving brain-damaged patients[33,86] and those employing transcranial magnetic stimulation (TMS) on the parietal cortex[28,29,80]. This more detailed insight contributes to a better understanding of the precise temporal mechanisms underpinning cognitive processes.

TRWs may be linked to another neural temporal phenomenon: oscillations. The relationship between the temporal integration hierarchy and oscillations is still unclear. A gradient of oscillatory frequencies, similar to the timescales gradient[52], has been described along the posterior-anterior cortical axis[87]. Gao and colleagues[52] suggested that the gradients of oscillations and neural receptive windows may (at least in part) share circuit mechanisms at different spatial scales, based on the similarity of these gradients and on known mechanisms of asynchronous and oscillatory population dynamics, analogous to the relationship between characteristic frequency and decay constant in a damped harmonic oscillator model. In the context of attention, theta rhythms from frontoparietal attentional networks have been proposed to rhythmically sample and temporally organize sensorimotor functions, creating alternating periods of attentional focus or shift[63,88]. Thus, conceptually, neural oscillations may serve as 'broadcasted' attentional signals affecting other brain regions. Similarly, TRWs can be thought of as 'receivers' of oscillatory attentional signals, determining how attentional modulation is processed. For example, the length of the TRW can determine how much of the oscillation's period will be summed together, thus generating a differential modulatory effect of the same oscillation frequency along different parts of the attentional gradient. Although we did not find evidence for the involvement of theta phase in the observed attentional effects, further research is needed to explore the relationship between these phenomena and test the hypothesis that they interact and influence each other along the attentional gradient and together dynamically contribute to attentional processing.

iEEG provides robust direct signals with unparalleled spatiotemporal resolution in humans, but it also has limitations[48–50]. Although contacts with epileptic activity are discarded from the analysis, iEEG data is collected from a pathological population, which might not be a valid model for neurotypical cognition. However, the fact that our participants demonstrated a neurotypical pattern of behavioral responses is reassuring in this respect. In addition, iEEG has a limited and inhomogeneous spatial coverage, determined solely by medical needs. We mitigated this limitation by collecting a large set of data from 28 patients, thus achieving a comprehensive coverage, and by considering the coverage in our analyses when needed, i.e., when comparing cluster hemispheric lateralization. As a result, some parts of the puzzle might be missing, yet the high signal-to-noise ratio and the

excellent resolution in the covered regions ensure that the activity recorded from them is robust.

Our findings challenge traditional attention models of multi-step processing across visual areas. They indicate that exogenous attentional effects follow a continuous neural trajectory across large-scale spatiotemporal gradients, where distinct processes of segregation and integration of attentional events occur. These neural dynamics provide the mechanisms through which the timing of attentional events shapes neural processing and consequently our behavior. Our findings suggest that the circuits for attention form a dynamic network, in which attentional effects are properties of the overall network, not separate functions assigned to different parts[89], and thus place exogenous attention processing in the context of the larger topographical organization of the human brain.

## Methods

### Participants and recordings

Thirty-one patients (aged $31.8 \pm 8.3$ years, 16 women; See Table 1 for full details) with drug-resistant focal epilepsy, hospitalized at the Pitié-Salpêtrière Hospital in Paris, participated in this study after giving their informed consent (CPP Paris VI, Pitié-Salpêtrière Hospital, INSERM C11–16). Three patients were excluded post hoc because of severe cognitive impairments and abnormally long response times (1 patient) or because of the presence of widespread brain lesions (2 patients), leaving a total of 28 included patients. For medical reasons, patients underwent intracerebral recordings by means of stereotactically implanted, multilead intracerebral depth electrodes (iEEG). Patients' experimental recordings were performed 4–14 days post-implantation, while their antiepileptic medication was gradually decreased and/or stopped. Patients were implanted with 5–12 platinum electrodes (AdTech®, Wisconsin) endowed with 4–12 contacts with a diameter of 1.12 mm and length of 2.41 mm, with nickel-chromium wiring. The distance between the centers of the two contacts is 5 mm. Electrode placement was uniquely determined by clinical criteria. In 13 patients, neuronal recordings were performed using an audio–video–EEG monitoring system (Micromed), which allowed simultaneous recording of 128 depth-EEG channels sampled at 1024 Hz (0.18–220 Hz bandwidth). In 18 patients, the recording was done with a Neuralynx system (ATLAS, Neuralynx, Inc.), allowing to record up to 160 depth-EEG channels sampled at 4 kHz (0.1–1000 Hz bandwidth). The least active electrode (preferably in white matter) was defined as the reference electrode. Before analysis, all signals were down-sampled to 512 Hz and re-referenced to their nearest neighbor on the same electrode, yielding a bipolar montage. Bipolar montage helps eliminate signal artifacts common to adjacent electrode contacts (such as 50 Hz line artifact) and achieves a high local specificity by canceling out

effects of distant sources that spread equally to both adjacent sites through volume conduction.

Spatial localization of the electrode was automatically computed in native space using the Epiloc toolbox[90] developed by the STIM engineering facility at the Paris Brain Institute (https://icm-institute.org/fen/cenir-stim//) using co-registered pre-implantation 1.5 T or 3 T MR scans and post-implantation CT scans. Each contact localization was automatically labeled according to the Desikan–Killiany–Tourville atlas parcellation[91] in patients' native space, using the Freesurfer image analysis suite (http://surfer.nmr.mgh.harvard.edu/) that is embedded in Epiloc. In 10 participants with low-quality MRI scans for which automatic contact labeling was not possible, two experimenters labeled manually and independently the contacts (inter-rater reliability $R = 0.99$) based on anatomical landmarks in the patient's native space, according to the parcellation of the Desikan–Killiany–Tourville atlas[91]. Contact localizations in standard MNI152 space were visualized with the BrainNet Viewer[92] Matlab toolbox (http://www.nitrc.org/projects/bnv/bnv/; Matlab R2016b and R2020a, The MathWorks, Inc.).

### Experimental task

A PC Dell Latitude D600 running E-prime 3.0 software (Psychology Software Tools, Pittsburgh, PA) controlled the presentation of stimuli, timing operations, and data collection. Stimuli were presented on a black background. Two empty gray boxes (3° long and 2.5° large) were horizontally arranged around a central fixation point located at the center of the screen. The distance between the center of the fixation point and the center of each box was 7.7°. The fixation point consisted of a gray plus sign (0.5° × 0.5°). Cues consisted of a 100-ms thickening (from 1 mm to 3 mm) of the contour of one lateral box. The target was a white "X" (1° in height), appearing at the center of one of the lateral boxes, with equal probability. Patients sat in front of the computer screen at a distance of approximately 57 cm. Figure 1A illustrates the experimental procedure. Each trial began with the appearance of the fixation point and the two placeholder boxes for 1000 ms. The cue followed for a duration of 100 ms. After a stimulus-onset asynchrony (SOA) of either 150 ms or 600 ms, the target appeared and remained visible for 150 ms. The placeholder boxes disappeared when a response was detected or after 3000 ms if no response was made. The experiment consisted of a total of 3 blocks of 112 trials, comprising 50 short SOA trials, 50 long SOA trials, and 12 catch trials, in which no target appeared after the cue, all randomly interleaved. Cues were non-informative, i.e., they indicated the target location on 50% of trials (Congruent location) and the opposite location (Incongruent location) on the remaining 50% of the trials. Patients were instructed to maintain their gaze at the central fixation point throughout the test and to respond to the target as fast and accurately as possible by pressing the right mouse button with their right index finger. The gaze position was verified by confrontation. The mouse was placed in an approximately central position with respect to the patient's body midline. It was stressed that the position of cues was useless for predicting the target position and should not be taken into account when responding. Before the first experimental block, patients performed 10 practice trials.

### Behavioral analysis

For each participant, trials with response time (RT) exceeding 3 std or faster than 100 ms were excluded from the analysis. Participants' mean RT was compared using a 2-way repeated measures ANOVA, with Congruence and SOA as factors, using JASP software (version 0.14.1)[93]. All post hoc comparisons were corrected for multiple comparisons using the Holm correction.

### iEEG preprocessing

Data preprocessing was done using the FieldTrip toolbox for EEG/MEG analysis (Donders Institute for Brain, Cognition and Behavior, Radboud University, the Netherlands. See http://fieldtriptoolbox.org)[94]

and Matlab (Matlab R2016b and R2020a, The MathWorks, Inc.). Continuous iEEG signals were visually inspected. Electrodes with excessive epileptic spikes located at or near the epileptic focus were rejected. Then, time windows showing epileptic transient activity were identified and excluded from further analysis. Next, epochs were extracted between 1 s before target onset and 1.5 s after target onset. Additionally, epochs were extracted between 1 s before the response time and 0.4 s after it. A second artifact rejection procedure was then performed on the epoched data, and trials with excessive variance, maximal signal, or kurtosis of their signal distribution were semi-automatically rejected. After epileptic artifact removal, 1403 of the bipolar contacts were usable for analysis, 671 of them were in the left hemisphere and 732 in the right hemisphere (see Fig. 2A and Table 2 for the localization of the usable contacts). According to the Desikan–Killiany–Tourville atlas parcellation[91], 336 (23.9%) of the contacts were located in the frontal lobe, 689 (49.1%) in the temporal lobe, 48 (3.4%) in the occipital lobe, 138 (9.8%) in the parietal lobe, 46 (3.2%) in subcortical regions and 146 (10.4%) in white matter.

A pseudo-whole-brain analysis approach was selected, focusing on high-frequency broadband (HFBB) activity (55–145 Hz a-priori range), a marker for multi-unit neural activity[95], which was associated with various cognitive processes[62,88]. HFBB power was extracted from each bipolar contact time series by convolving the signal with a set of complex Morlet wavelets (with 8 cycles), in 20 logarithmically spaced center frequency bands. Every trace was separately baseline-corrected by means of a z-score relative to the trials' baseline distribution in the 700 ms prior to cue onset, separately for each of the frequency bands. This approach accounts for the $1/f$ signal drop-off in the high-frequency band with increasing frequencies. Finally, we discarded the edges to avoid filter artifacts and extracted individual non-overlapping trials relative to either target onset (−0.9 to 1.36 s) or relative to the response time (−0.9 to 0.3 s). HFBB signals were down-sampled to 50 Hz for further analysis.

### Trajectory k-means clustering

In order to reveal contacts' prototypical temporal patterns of activity across experimental conditions, we developed a custom-made clustering approach based on k-means clustering, implemented through Matlab (Matlab R2016b and R2020a, The MathWorks, Inc.). Clustering was done on responsive contacts, defined as having a target-locked significant effect ($p \leq 0.05$ uncorrected) of at least 100 ms in one or more of the eight experimental conditions compared to baseline. For each condition in a given contact, a time-resolved independent samples t-test was performed, in which each time point across trials was compared to the distribution of all the baseline samples pooled over all that condition's trials (−0.2 to 0 s prior to cue onset). This yielded 644 contacts (See Table 2 for their spatial localization), and their mean target-locked or response-locked activity time series were transformed into an 8D matrix, where each dimension corresponded to one of the eight experimental conditions (short/long SOA × congruent/incongruent × contralateral/ipsilateral target relative to the recording contact; see Fig. S2A, B for an illustration and example). The trajectories, consisting of the mean target-locked or response-locked HFBB power across the 8-dimensional condition space, were entered into the clustering algorithm. Activity across conditions was z-scored relative to the distribution of the trials' entire duration. Trajectories were iteratively partitioned (10,000 iterations) into 2–9 clusters, in which each contact was assigned to the cluster with the nearest centroid trajectory. This was achieved by minimizing the sum of the Manhattan distances, time-point-by-time point to quantify trajectory similarity while preserving temporal order. Based on the elbow method[96] the 6-cluster solution was chosen for the clustering of target-locked activity (see Fig. S2). Figure S6 shows the clustering of target-locked activity for 2–8 cluster solutions, demonstrating the stability across different k solutions of the three clusters further analyzed. The stability

was assessed using contingency tables analysis performed using JASP[93], estimating the correspondence between the contacts assigned to these three clusters and specific clusters from each k solution. There was a strong significant correspondence between the assignment of contacts to clusters in the 6-cluster solution and in the other k solutions (Table S1). A k-solution cluster was marked as stable if the main group of contacts composing it could be mapped to one of the three further analyzed clusters, which in turn shared most of its contacts with that cluster (Fig S6, Table S1). Based on the elbow method[96], for the clustering of response-locked activity, a 7-cluster solution was chosen (See Fig. S4). In order to identify the correspondence between target-locked and response-locked clusters, a contingency tables analysis was performed using JASP[93]. The distribution of the 28 participants' contacts across target-locked and response-locked clusters is shown in Fig. S4, demonstrating that clusters did not result from any single participant's temporal activity but rather reflected temporal patterns across many participants. The linear correlation between the centroid time series of all conditions across target-locked clusters revealed that out of the six target-locked clusters, three had a dynamic temporal profile across the different experimental conditions. These clusters were positively correlated among themselves, forming a distinct cluster group (see Fig. S5). The correlation pattern within the remaining three clusters was more uniform and negatively correlated across clusters. Clusters 1, 2, and 3 were used as a type of functional region of interest for further analyses. We chose to focus on these clusters because of their stability across clustering solutions and their variable responses across experimental conditions (Fig. S5). Conversely, even if the remaining clusters might contribute to the processing of the different attentional conditions, they could not explain the differences between them, given that their correlation pattern across experimental conditions was uniform (Fig. S5).

### Cluster hemispheric lateralization
The hemispheric lateralization of the clusters was tested on a subgroup of contacts localized in cortical volumes that were sampled in both hemispheres. This was done to overcome the confound of unequal coverage within the hemispheres. To identify similarly-covered contacts, a 3 mm radius sphere (corresponding to the assumed volume recorded by iEEG contacts[50]) was fit around each contact using SPM12[97], and the overlap between each of the spheres and the entire covered volume in the other hemisphere was calculated. The cluster distribution of the 309 resulting contacts (148 in the left hemisphere and 161 in the right hemisphere) across the hemispheres was compared using a contingency table analysis in JASP[93], and post hoc binomial test with Holm correction was conducted to identify the clusters with significant hemispheric lateralization.

### iEEG statistical analyses
All statistical analyses were performed using statistical toolbox in Matlab (Matlab, R2020a, The MathWorks, Inc.) and JASP version 0.14.1[93].

**IOR-related neural activity.** In order to test which of the cluster's neural activity was IOR-related, we compared Congruent and Incongruent trials in the long SOA condition, For each cluster, we performed a time-resolved 3-way ANOVA (Fig. 4) with Congruence, Contact's Hemisphere and Target Laterality (relative to the contact), on the target-locked HFBB signal in each time point (between 0 and 0.8 s post target onset), across all the cluster's trials (pooled over contacts and participants). Holm multiple comparisons correction was applied over all the time points within each main effect and interaction. Post hoc comparisons were performed on time points in which the Congruence x Hemisphere interaction was significant, with Holm correction for multiple comparisons. Detailed ANOVA-corrected p-values for each cluster are shown in Table S2.

**RT-modulation of target-locked neural activity and visual modulation of response-locked neural activity.** In order to test which of the clusters' neural activity was modulated by the RT, we sorted in each cluster all the trials pooled over the conditions according to their RT. We then binned them into 20 quantiles (Fig. S8A). Within each cluster, we tested the effect of the RT-bin using a time-resolved 1-way repeated measures ANOVA, on mean target-locked HFBB signal across conditions, in each time point (between 0 and 0.8 s post target onset; pooled over contacts and participants). Holm multiple comparisons correction was applied over all the time points. A similar analysis was performed on the response-locked clusters. Because RT is defined as the time from target onset to the response, this procedure sorted the response-locked trials according to target onset and thus could unveil visual modulation of response-locked activity.

**Temporal gradient analysis.** Within each target-locked cluster, the contacts' time of the maximal HFBB power (between 0 and 0.6 s post target onset) was identified separately for Congruent and Incongruent long SOA conditions. Contacts' peak times were compared across the three clusters using a mixed-repeated measures ANOVA, with Congruence as a within-subjects factor and Clusters as a between-subjects factor. A linear post-hoc polynomial contrast was used to test if peak time was linearly ordered across clusters. A similar analysis was performed on the response-locked clusters.

**Core–periphery gradient analysis.** In order to test if the clusters' anatomical localization followed the core–periphery gradients, the MNI coordinates of target-locked clusters' contacts were assigned the closest voxel's gradient value on the two principle gradients described by Margulies et al.[53]. The distances between contacts and the closest voxels did not differ across clusters (1-way ANOVA, $F_{(2,230)} = 0.064$, $p = 0.94$). Contacts' gradients' values along the two gradients were compared using a 1-way ANOVA with Clusters as a factor. A linear post-hoc polynomial contrast was used to test if clusters were linearly ordered along the two gradients. A similar analysis was performed on the response-locked clusters. Here too, the distances between contacts and the closest voxels did not differ across clusters (1-way ANOVA, $F_{(3,246)} = 1.23$, $p = 0.30$).

**Estimation of temporal receptive window length.** TRW length was assessed by computing the across-trial autocorrelation[66,98] of the non-filtered iEEG signal (down-sampled to 100 Hz; 350–1150 ms post target), for each of the contacts in the three target-locked clusters. An exponential decay function ($e^{(-t/\tau)}$) was fit to the contacts autocorrelation coefficient across time lags. TRW length for each contact was defined as the time constant ($\tau$) of the contact's fitted exponential decay function, i.e., the time it takes for the autocorrelation to decrease by a factor of $e$[66,98].

### Structural connectivity of pre and post-rolandic contacts
To determine the connectional anatomy of the three clusters, we used fiber tracking in a sample of 176 healthy controls from the Human Connectome Project database[59] and used a threshold-free cluster enhancement (TFCE)-based non-parametric t-test to determine the significant tracts. Contacts of each cluster were fitted with a 3 mm radius sphere around them as described above, and labeled as pre or post rolandic, using the central sulcus as a reference point in patient's native space (Number of pre and post rolandic contacts per cluster: Cluster 1–8:60, Cluster 2–23:74, Cluster 3–34:33). The resulting pre and post rolandic contact spheres were used as region-of-interests (ROIs) to identify white matter fibers connecting them. This fiber-tracking analysis was done on the high-resolution 7 T MRI scans of 176 healthy individuals from the Human Connectome Project database[59] using TrackVis 0.6.1 (http://trackvis.org/). The resulting tractography maps were binarized, and significant tracts across individuals were

determined using a threshold-free cluster enhancement (TFCE)-based non-parametric t-test in FSL 6.0 (1000 permutations, height threshold of 0.95 to control significance level at $p < 0.05$; https://fsl.fmrib.ox.ac.uk/fsl/fslwiki/FSL). The corrected t-maps were then used to identify the number of white matter voxels that overlapped with the SLF tracts templates of the white-matter probability maps of the BCBtoolkit (http://toolkit.bcblab.com/). In order to identify the tracks overlapping with the three branches of the SLF, probability maps were thresholded at 50%, yet the large overlap between the tracts of SLF II and SLF III templates (present even with a 90% probability threshold) made the differentiation between them difficult. The number of significant overlapping voxels between corrected t-maps and SLF maps was calculated per hemisphere. The corresponding voxels were then normalized for the number of significant voxels in the corrected t-maps [(Nr of overlapping voxels per SLF tract/ Nr of significant voxels in the corrected t-maps in the respective hemisphere) *100].

### Reporting summary

Further information on research design is available in the Nature Portfolio Reporting Summary linked to this article.

## Data availability

Raw iEEG and patients' MRI and CT data cannot be shared due to ethics committee restrictions. Intermediate as well as final processed data that support the findings of this study are available from the corresponding author (T.S.M.) upon request. The diffusion MRI data used in this study are available in the HCP database https://www.humanconnectome.org/study/hcp-young-adult/document/1200-subjects-data-release.

## Code availability

The custom codes used to generate the figures and statistics are available from the lead contact (T.S.M.) upon request.

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

## Acknowledgements

We would like to thank Pietro Avanzini, Danilo Bzdok, Florence Bouhali, and the PICNIC lab at the Paris Brain Institute (ICM) for invaluable discussions and assistance. This work was funded by Israel Science Foundation postdoctoral fellowship number 57/15 (T.S.M.), Marie Sklodowska Curie fellowship 702577-DynamAtt (T.S.M.), and Agence Nationale de la Recherche BRANDY grant ANR-16-CE37-0005 (T.S.M., D.J.B., V.N., S.F.V., P.B.).

## Author contributions

Conceptualization: T.S.M., D.J.B., P.B., Data curation: K.L., Formal analysis: T.S.M., Methodology: T.S.M., J.D.S., B.C.K., J.L., Investigation: T.S.M., D.J.B., Visualization: T.S.M., B.C.K., Funding acquisition: T.S.M., P.B., Project administration: T.S.M., Resources: K.L., S.F.V., V.N., C.A., V.L., D.S.M., Software: T.S.M., D.J.B., A.B., J.D.S., S.F.V., D.S.M., Supervision: P.B., Writing—original draft: TSM, Writing—review & editing: T.S.M., D.J.B., B.C.K., A.B., K.L., S.F.V., V.N., C.A., V.L., D.S.M., J.D.S., P.B.

## Competing interests

The authors declare no competing interests.
