## [Peer Review File · Nature Communications]

Intracortical recordings reveal Vision-to-Action cortical gradients driving human exogenous attentionREVIEWER COMMENTS

Reviewer #1 (Remarks to the Author):

Summary: This study uses intracortical recordings implanted in 28 individuals to examine how exogenous attention modulates processing dynamics in the human brain. An exogenous cuing paradigm (Posner cuing) was used to induce attentional orienting at two different temporal windows – short (150ms) and long (650ms). It was observed that timing, location, and task-relevance of attentional events defined a spatiotemporal gradient of three neural clusters. It is argued that these findings reveal how the psychological construct of exogenous attention emerges over large-scale cortical gradients.

The project is ambitious, and I commend the authors on executing what appears to be an incredibly labor-intensive project. The paper is well written, and the methods are appropriate. I am not an expert on intracranial recordings though, so I anticipate other reviewer(s) will cover this part.

There are a few points that the authors may wish to consider. In short: I would like to see clarifications on the paradigm that was used, a revised interpretation of the behavioral data and how that influences interpretations of neural signals in the three major neural clusters, and an enhanced motivation and theoretical interpretation.

1. Introduction – the introduction does a good job of introducing exogenous attention and the corresponding neural network that is associated with it (TPJ and FEF). It also draws a distinction between exogenous attention facilitation and IOR. However, it is not well laid out as to how the current investigation by employing intracranial recordings will contribute to our understanding of current theories of attentional selection. Bottom line is that introduction reads entirely as atheoretical, without predictions according to any particular hypothesis or theory.

2. Method – Another major concern has to do with the fact that the cue manipulation did not actually yield the expected/desired behavioral effect. If exogenous attention were to be successfully measured/manipulated then we would see a behavioral benefit for congruent trials over the incongruent trials at short SOA – faster RTs for congruent. This was not observed, and it is worrisome. Do the authors have an interpretation as to why the short SOA manipulation did not yield the expected finding?

Related to this. It is stated that catch trials were used however no description is given as to whether catch trials were analyzed and what the false alarm rate was for these trials. It might be that some of the participants got “caught.” You might also be able to filter your data according to high catch rates and in that case be able to reveal facilitatory effects.

It is difficult to talk about exogenous attention without seeing an effect of facilitation. This at least needs to be thoroughly discussed. The neural data interpretation hinge on this.

3. In several parts of the manuscript, as it relates to Cluster 3 response, the authors talk about how effects are related to motor responses – stronger activation for response-requiring targets than for cues in the “left hemisphere, contralateral to the responding hand.” However, motor responses were not

manipulated, unless I missed it. The only response hand was the right one. Thus it is unclear whether the left hemisphere response is simply stronger or whether it is related to the response hand. Please clarify, and perhaps modify interpretation accordingly.

4. From Discussion:

“Our results suggest that this cluster’s activity represents a key attentional processing, associating perception and action signals.” This is a conclusion statement yet it is vague. It is unclear what key attentional processing is referred to here.

“TPJ contacts were activated also in response to cues, and in congruent trials, when target location corresponded to the location of the preceding cue, therefore suggesting that the TPJ is not just a circuit breaker responding when unexpected and pertinent targets appear and reorienting of attention is needed” – I am not sure I follow this. Cues are salient events that attract exogenous attention independent of whether the target that follows it is congruent with the cue or incongruent.

Reviewer #2 (Remarks to the Author):

In the present manuscript, the authors study the neurophysiological correlates of exogenous attention using a large dataset of intracranial recordings from the human brain. They employ a well-established classic attention task that has first been reported by Posner et al. in the 1980s.

On the behavioral level, they replicate the finding of inhibition-of-return, albeit only two time points (150ms and 600ms) were probed. On the neural level, they leverage the power of intracranial recordings in humans and extract high-frequency band activity as a proxy of multiunit spiking. Employing a clustering approach, they identify three key clusters spanning sensory, association and motor areas, where only the cluster in the association areas displays a condition-specific difference.

Overall, this manuscript is rather dense in terms of methods, the authors introduce e.g. a multivariate trajectory approach to map condition differences to neural data and employ a variety of data-driven cut-offs to determine the optimal cluster size. Finally, they also incorporate the concept of intrinsic time scales and fiber tracking.

In its present form, the manuscript describes behavior and neural data separately and it remains unclear if the authors truly identified an underlying mechanism that explains the behavioral effect. The

manuscript mainly requires several clarifications in terms of methods and might benefit from clearly laying out the overall analytical approach.

Overall, the manuscript seems a bit convoluted, since when ignoring the methods-details, one is left with (a) replication of the behavioral IOR (albeit no facilitation at the early interval), (b) identification/replication (based on functional rather than anatomical data) of the well-known hierarchy sensory->association cortex->motor, (c) the new finding that the IOR is mainly mediated at the level of association areas and (d) that the identified hierarchy tracks the timescale hierarchy (as e.g. identified recently, Gao et al. 2020 eLife). While the paper obviously constitutes a tour-de-force in terms of methods and rather dense, it lacks a clear take-home message of what the key finding is that advances the current conceptual scope. In the title, the authors suggest that the gradient per se might reflect the behavioral correlate, but that seems not substantiated in the manuscript.

Furthermore, the authors loosely refer to the concept of rhythmic attention, as recently introduced by e.g. Landau et al. or Fiebelkorn and Kastner. This idea remains a bit unevaluated in the paper: (a) Could it be that the behavioral effect is solely driven by theta-phase dependence (when assuming a 6-7Hz rhythm, then activity at 150ms and 600ms would fall into opposite phase bins, hence, maximizing the behavioral contrast) and (b) the relationship of the temporal hierarchy and the rhythmic sampling (which both constitute some sort of behavioral time signature) should at least be addressed in the discussion to highlight that both phenomena are most likely distinct entities.

Taken together, this is an impressive manuscript that is methods-rich and provides several novel ideas how to analyze complex human intracranial data. The link to behavior could be substantiated and the rationale could be better motivated. In addition, several queries remain, mainly to unpack the manuscript and better guide the reader through the various steps and conclusions.

1. The authors employ a variety of methods, it would be beneficial if the analysis pipeline could be laid out in the beginning, with a clear rationale why the approaches were chosen. The 8D condition space analysis for instance, on which several subsequent analyses rest, requires a bit more of an illustration and explanation. Ideally, the authors might consider modeling the different possible outcomes to give the reader a clearer intuition of the method. From solely reading the description without any visual depiction, it seems that the 8D space is mainly composed of binary (left or right) values. Furthermore, it remains unclear how these trajectories based on behavioral data were linked to EEG activity.

2. In the main clusters, the authors report only very few electrodes (approx. 200 out of a large sample), raising the question what happens at all the other electrodes. Is there truly no effect or modulation? This might also serve as a baseline condition. To me, the link between the three main clusters and the late activation/suppression is unclear. Are these the results of two different analysis or are they the result of

the very same analysis? In Figure S3 different clusters are marked in color – is this the outcome of a test or did the authors draw these boxes based on visual inspection?

3. Figure 2 is labeled as ‘IOR-related neural activity’. Unfortunately, I fail to observe the direct correlate of the behavioral effect with similar activity/behavior at the short interval and a dissociation at the longer interval. Likewise, Figure S7 and S9 seems to show behavioral correlations in all clusters, which is different to their assertions in the main text, which highlights the difference in association cortex.

4. Figure 4: Is there any statistical quantification of the fiber tracking? After all it seems that this was obtained in a different dataset.

5. Is the ACW/time scale related to the behavioral effect? Or is this just a secondary observation that is only loosely tied to the main results? In other words, is the change in ACW epiphenomenal and solely a result of clustering activity along the hierarchy, or is there something specific about the timescale in association areas that explains the IOR?

6. Later in the manuscript, the authors split cluster 2 into cluster 2a and 2b and the rationale and functional criteria to do so remain unaddressed.

In sum, this is an interesting manuscript that describes novel electrophysiological data that might explain a well-established behavioral phenomenon. In its present form, the manuscript would benefit from a separate paragraph of overall analysis strategy, that in particular needs to motivate the cluster approach and the link of the 8D behavioral matrix to the EEG analysis. Also, the authors need to highlight if the ACW and fiber tracking analysis is only the consequence of the clustering or if they have evidence for assigning a true functional role. Currently, both are presented as an afterthought. In sum, the manuscript is dense and a bit convoluted, but the overall data and analysis strategy is interesting. Given the multitude of analyses, evaluating the key statistics is difficult. Including effect sizes for stats would help. Also, based on binomial distributions, the authors might want to assess how many electrodes from the entire sample need to show an effect in order to deem it significant. Currently, some clusters contain very few electrodes, raising the question if this would fall within the expected noise distribution.

Overall, this constitutes an impressive tour-de-force, but given its density the proposed mechanistic link between behavior and large-scale organization remains a bit difficult to understand.

REVIEWER COMMENTS

Reviewer #1 (Remarks to the Author):

Summary: This study uses intracortical recordings implanted in 28 individuals to examine how exogenous attention modulates processing dynamics in the human brain. An exogenous cuing paradigm (Posner cuing) was used to induce attentional orienting at two different temporal windows – short (150ms) and long (650ms). It was observed that timing, location, and task-relevance of attentional events defined a spatiotemporal gradient of three neural clusters. It is argued that these findings reveal how the psychological construct of exogenous attention emerges over large-scale cortical gradients. The project is ambitious, and I commend the authors on executing what appears to be an incredibly labor-intensive project. The paper is well written, and the methods are appropriate. I am not an expert on intracranial recordings though, so I anticipate other reviewer(s) will cover this part. There are a few points that the authors may wish to consider. In short: I would like to see clarifications on the paradigm that was used, a revised interpretation of the behavioral data and how that influences interpretations of neural signals in the three major neural clusters, and an enhanced motivation and theoretical interpretation.

We thank the Reviewer for their positive assessment. In the revised manuscript, we have now clarified the paradigm and our interpretation of the behavioral and neural results, as well as the motivation and theoretical approach to this project, as detailed below.

1. Introduction – the introduction does a good job of introducing exogenous attention and the corresponding neural network that is associated with it (TPJ and FEF). It also draws a distinction between exogenous attention facilitation and IOR. However, it is not well laid out as to how the current investigation by employing intracranial recordings will contribute to our understanding of current theories of attentional selection. Bottom line is that introduction reads entirely as a theoretical, without predictions according to any particular hypothesis or theory.

Reply:

We thank the reviewer for the positive feedback regarding the Introduction.

Our motivation for using human intracerebral recordings was to address the gap in our current understanding of the neural basis of exogenous attention, particularly in relation to two hallmark attention-associated behavioral phenomena: RT facilitation and IOR. Despite decades of research, the evidence supporting each of the contentious neural theories attempting to explain these phenomena is limited, indirect, and often contradictory. While some theories propose that IOR is caused by attentional capture of previously cued locations or an inhibitory attentional bias, others suggest it may occur due to sensory adaptation, habituation, and/or caused by motor/decisional biases. These theories are still subject to significant debate, and the evidence supporting each theory remains inconclusive (Lupiañez et al., 2006; Martín-Arévalo et al., 2016).

The lack of consensus on the neural basis of exogenous attention is partly due to the fact that prior work investigating the neural basis of these fast and dynamic processes is quite sparse and has been based on non-human primates or indirect human neuroimaging methods with limited spatial or temporal resolution. To address these issues, we used human intracerebral recordings, which allowed us to track human attentional dynamics directly with high temporal resolution and excellent spatial precision over large brain topographies. These advantages are critical when examining the neural localisation and computations of exogenous attention.

We chose to use a data-driven approach that leverages the high spatiotemporal resolution of iEEG to establish the neural dynamics associated with the perceptual, attentional, and motor aspects of the Posner task and to test whether the data converge with existing theoretical frameworks. By doing so, we aimed to provide new insights into the neural basis of exogenous attention and to clarify the existing theoretical debate surrounding RT facilitation and IOR.

In the revised manuscript, we added the following paragraphs to the introduction to better motivate our approach, and clarify its positioning relative to the current theoretical background:

There are several contentious neural theories of IOR, but very few about RT facilitation and the evidence supporting each of them is limited, indirect and often contradictory. Theories of IOR diverge on the mechanistic nature of IOR and its putative localization(s) in the brain (sensory/attentional and/or motor/decisional). It was suggested, for instance, that IOR is caused by attentional capture of previously cued locations³⁷, perhaps by delaying bottom-up signals of the salience map^{26,27,38}, or by an inhibitory attentional bias^{39,40}. A recent theoretical model based on the known architecture of frontoparietal cortical networks and on their anatomical and functional asymmetries⁴¹, proposed that IOR, that arises from a noise-increasing reverberation propagation of activity within priority maps of the frontoparietal circuit linking frontal eye field (FEF) and intraparietal sulcus (IPS)⁴². Other theories proposed that IOR might occur early, over perceptual neural pathways through the reduction of stimulus salience around a previously attended location⁴³, or due to sensory adaptation⁴⁴ or habituation⁴⁵. IOR was suggested to occur also later in processing, involving motor/decision circuits, in the form of a bias against responses toward previously attended spatial locations⁴³, motor habituation⁴⁵ or an oculomotor activation signal⁴⁶. For example, the Cue-target event integration-segregation hypothesis⁴⁷ postulates that the summation of early and late perceptual processes, spatial selection processes and decision processes, determines together if the net behavioral effect is facilitatory (RT facilitation) or inhibitory (IOR)^{6,11}. According to this theory, binding together sequential stimuli that share similar features (such as location and close-timing) into a single event file⁴⁸ can lead to facilitatory effects helping to select the target location in advance⁶. However, binding can also cause inhibitory effects when the similar sequential stimulus needs to be detected as a new separate event, resulting in a cost in detecting the onset of the target⁶. These theories remain highly debated, and the evidence supporting each one is inconclusive. This is due at least in part to the fact that prior work investigating the neural basis of these fast and dynamic processes is quite sparse, and based either on high-resolution recordings in specific brain regions in non-human primates, or on indirect human neuroimaging methods with limited spatial resolution, such as EEG, or with limited temporal resolution, such as functional MRI. These considerations are critical when studying the neural correlates of exogenous attention, which operates on a very rapid time scale and dynamically involves large neural networks over the entire brain, thus rendering past findings not informative enough for supporting or refuting existing neural theories of

attention. Thus, our understanding of these attention processes stays fragmented, leaving the involved networks and underlying mechanism obscure.

Our aim here was to establish the large-scale spatiotemporal neural dynamics of the mechanisms involved in the exogenous orienting of spatial attention. We chose to use intracortical EEG (iEEG) in humans^{37–39}, acquired across 28 patients (1403 contacts), to achieve comprehensive cortical coverage. iEEG is the only method that allows to track human attentional dynamics directly (i.e. invasively) with high temporal resolution and excellent spatial precision over large brain topographies, crucial for capturing rapid attentional dynamics across the brain. Due to the lack of consensus on the neural basis of exogenous attention, we opted for a data-driven approach, leveraging the advantages of iEEG to establish how neural activity tracks the visual, attentional and response aspects of the classic Posner exogenous attention task⁷ and to test whether the findings converge with existing theoretical frameworks. This approach allowed us to study the impact of attentional cues on the detection of subsequent targets, as a function of the delay between them.

Our revised discussion now focuses on how our findings support the Cue-target event integration-segregation hypothesis.

Page 31: Correspondingly, our finding that TRWs lengthen along the cluster gradient reveal potential temporal operations at the basis of exogenous attention. Furthermore, the integration of cue-target responses in Clusters 2 and 3 in the long-SOA could reflect temporal pooling⁵⁹. In Cluster 1, situated lower on the gradient, TRWs are shorter, allowing for segregation of activity even at short delays. In upstream frontoparietal clusters where TRWs are longer, cue- and target-induced responses resulted in a single activity peak. This temporal pooling might group the cue and target in a single event⁸⁰, leading to RT facilitation at short cue-target delays^{6,42,80}. These findings dovetail with the hypothesis that RT facilitation results from a summation of cue-related and target-related responses, thus reflecting hard-wired limitations of the neural system that cannot respond separately to rapidly repeated stimuli, and processes them as a single event^{6,42,80}. Longer cue-target delays could instead provide the system with enough time to segregate cue- and target-related responses^{6,42}.

In the revised Discussion (Page 31) we expanded this point:

According to Cue-target event integration-segregation hypothesis⁶ RT facilitation arises when the net effect of facilitatory processes, such as exogenous spatial attention orienting and binding-associated spatial selection benefit, is larger than the detection cost the binding might cause due to the difficulty to detect the onset of the second bound stimulus.

2. Method – Another major concern has to do with the fact that the cue manipulation did not actually yield the expected/desired behavioral effect. If exogenous attention were to be successfully measured/manipulated then we would see a behavioral benefit for congruent trials over the incongruent trials at short SOA – faster RTs for congruent. This was not observed, and it is worrisome. Do the authors have an interpretation as to why the short SOA manipulation did not yield the expected finding?

Reply:

We agree with the Reviewer that this is an important point. However, the absence of a behavioral effect in the short SOA is a common finding, which does not affect our main results regarding the neural basis of exogenous attention. Both RT facilitation and IOR are classic exogenous attention phenomena, each standing on its own, and studying each one of them, independently of the other, can contribute to establishing the neural basis of exogenous attention. Specifically, RT facilitation is an elusive effect, easily masked by other processes (Chica, Martín-Arévalo, et al., 2014; Lupiáñez, 2010; Martín-Arévalo et al., 2016). Our design was not optimal for unmasking the behavioral effect because of the lack of temporal overlap between cue and target, which is one of the conditions that favors the appearance of facilitation in detection tasks (Chica, Martín-Arévalo, et al., 2014). In addition, the Poffenberger effect we observed, i.e. faster RTs when visual stimuli appear on the same side as the responding hand compared to when they appear on the opposite side, further masked the RT facilitation effect. This was explained in the Results section on page 4 and Fig. S1 of the manuscript:

...facilitation at short-SOA failed to reach significance ($p=0.37$; see Fig. S1 for individual RT effects and target-side analysis), as is often the case with this subtle effect⁸. Moreover, left target Congruent RTs were slower than right target Congruent RTs, across both SOAs (Fig. S1B; repeated-measures 3-way ANOVA: Target-side X Congruence interaction- $F_{(1,27)}=8.28$, $p=0.008$, $\eta^2=0.007$), reflecting a Poffenberger effect^{54,55}, i.e. faster RTs for right cue & target than for left cue & target, when responding with the right hand. In Incongruent trials in which cue & target appear at opposite sides of the screen, this effect might have averaged out.

Yet, current theories postulate that even when difficult to observe behaviourally, the facilitation effect nevertheless exists (Lupiáñez, 2010). In our neural data in Cluster 2 in the short SOA, there is a combination of stronger responses for contralateral stimuli (Cue or Target) which are summed together, leading to a faster and stronger activation for contralateral congruent compared to contralateral incongruent cues and targets, and compared to ipsilateral ones (Significant Congruence X Target-side interaction effect, at -0.06-0.16 sec post target, see Fig 1 below). This differential summed activity translates to a neural preference for stimuli repeating in the same specific spatial (contralateral) location, dovetailing the behavioral RT facilitation effect, in which RT is faster for repeated stimuli in a specific location.

However, as a rule, we believe that analyzing neural data not supported by significant behavioral effects should be avoided as they are much harder to interpret. Yet, assuming that the facilitation effect is only masked behaviorally, the effect may still be observed neurally. Because this effect is much harder to interpret than the IOR neural effect observed in the same cluster, to comply with the Reviewer's comment we decided to cautiously report this analysis in the revised manuscript.

We added these findings to the revised Results (page 16):

Despite the lack of a significant behavioral effect of RT facilitation, the effect might be masked by other processes, as is often the case^{6,8,11}. Current theories postulate that even when masked, the facilitation effect nevertheless exists^{6,8,11}. We therefore performed an exploratory time-resolved ANOVA analysis with the factors Congruence, Target-side and Hemisphere, to test the attentional neural effect in Cluster 2 also in the short-SOA. In the target time-window, cluster 2 showed a significant Congruence X Target-side interaction effect (-60-140ms post target; largest $p=0.022$) and a main Target-side effect (160-300ms; 320-360ms; 440-

460ms post Target onset; largest $p=0.012$; see Fig. S7). This reflects a combination of stronger responses for contralateral stimuli (Cue or Target) which are summed together, leading to a faster and stronger activation for contralateral congruent compared to contralateral incongruent cues and targets, and compared to ipsilateral ones. This differential summed activity translates to a neural preference for stimuli repeating in the same specific spatial (contralateral) location, dovetailing the behavioral RT facilitation effect, in which RT is faster for repeated stimuli in a specific location.

We also expanded our discussion of the absence of a behavioral RT facilitation effect (Page 29):

Clusters 2 and 3 are located closer to core regions of the cortical gradient, and overlap with known frontoparietal attention networks^{19,67,68}. The neural activity in Cluster 2, occurring midway along the gradient, is sensitive to cue-target spatial positions and delays, and exhibits IOR-related onset and offset. Both visual processing of the target and manual response preparation shape the neural activity in this cluster, which is lateralized to the right hemisphere, consistent with lesion and neurostimulation data on IOR^{28–30,33,34}. Despite the fact that we did not find a significant behavioral effect of RT facilitation, the involvement of Cluster 2 neural activity in attentional computation in the short-SOA condition remains plausible. First, RT facilitation is an elusive effect, easily masked by other processes^{6,8,11}. Our design was not optimal for unmasking the behavioral effect because of the lack of temporal overlap between cue and target, which is one of the conditions that favors the appearance of RT facilitation in detection tasks⁸. In addition, the Poffenberger effect we observed further masked the RT facilitation effect. Yet, current theories postulate that even when difficult to observe behaviorally, the facilitation effect nevertheless exists^{6,11}. Our exploratory analysis revealed in Cluster 2 at the short-SOA a differential cue-target summed activity, which translates to a neural preference for stimuli repeating in the same specific spatial contralateral location. This neural effect dovetails the behavioral RT facilitation effect, in which RT is faster for repeated stimuli in a specific location. Therefore, our results suggest that this cluster's activity represents a key attentional processing for both short and long SOAs, associating perception and action signals.

Figure 1. Exploratory analysis of short-SOA congruence-related neural activity in the target time-window in Cluster 2. Mean target-locked short-SOA activity in Cluster 2 (red), computed over trials pooled across all cluster contacts, for Congruent trials (full lines) and Incongruent trials (dashed lines), when targets were ipsilateral (left) or contralateral to the recording contact (right). Note, that when targets were ipsilateral, Incongruent cues were contralateral (dark blue arrow), and Congruent cues were ipsilateral (light blue arrow), and conversely for contralateral targets. Responses to contralateral cues and targets were stronger than to ipsilateral ones, as shown by a significant Target-side x Congruence effect (shaded light red areas; -60-140ms post Target onset; largest $p=0.022$) and a main Target-side effect (shaded dark red areas; 160-300ms; 320-360ms; 440-460ms post Target onset; largest $p=0.012$). Shaded areas around traces depict SEM; Dashed vertical lines represent Target onset (black) and Cue onset (blue).

Related to this. It is stated that catch trials were used however no description is given as to whether catch trials were analyzed and what the false alarm rate was for these trials. It might be that some of the participants got “caught.” You might also be able to filter your data according to high catch rates and in that case be able to reveal facilitatory effects.

Reply:

We apologize for this omission. There were only 12 catch trials in total, each with either left or right cues. This small number is insufficient for reliable analysis. Importantly, participants never responded to catch trials (they were never “caught”). We added this information to the revised Results section (Page 4):

Catch trials were not statistically analyzed because of their small number, but patients never responded in these trials.

It is difficult to talk about exogenous attention without seeing an effect of facilitation. This at least needs to be thoroughly discussed. The neural data interpretation hinge on this.

Although we agree with the Reviewer that this point merits a more extensive discussion, we do not think that all data interpretation hinges upon it. As mentioned above, the absence of a behavioral effect in the short SOA is a common result, which does not affect our main findings regarding the neural basis of

exogenous attention. First, both RT facilitation and IOR are exogenous attention phenomena, each in its own right, and studying each one of them, independently of the other, can contribute to establishing the neural basis of exogenous attention. Moreover, the manipulation of the cue in the long SOA is an equally valid exogenous attention manipulation that yields more robust results. Consequently, we chose to focus on it. Finally, as mentioned above in response to comment #2, current theories postulate that even when difficult to observe behaviourally, the facilitation effect nevertheless exists (Lupiáñez, 2010). Our analysis following the Reviewer's comments showed that in Cluster 2 in the short SOA, there is a combination of stronger summed responses for contralateral stimuli, leading to a faster and stronger activation for contralateral congruent when compared to contralateral incongruent cues and targets, as well as in comparison to ipsilateral ones. This differential summed activity translates into a neural preference for stimuli repeating in the same specific spatial (contralateral) location, aligning with the classic behavioral RT facilitation effect, where RT is faster for repeated stimuli in a specific location.

3. In several parts of the manuscript, as it relates to Cluster 3 response, the authors talk about how effects are related to motor responses – stronger activation for response-requiring targets than for cues in the “left hemisphere, contralateral to the responding hand.” However, motor responses were not manipulated, unless I missed it. The only response hand was the right one. Thus it is unclear whether the left hemisphere response is simply stronger or whether it is related to the response hand. Please clarify, and perhaps modify interpretation accordingly.

Reply:

The Reviewer is correct, responses were made exclusively using the right hand to mitigate spatial compatibility RT effects arising from congruence between the responding hand and the side of the presented target. While we cannot entirely dismiss the possibility that the left hemisphere response is simply stronger, our interpretation is grounded in a comprehensive line of evidence, rather than relying solely on the left hemisphere asymmetry exhibited by Cluster 3 contacts. Importantly, this Cluster contains contacts within the right hemisphere as well, and these contacts are also situated in non-motor regions, including the posterior temporal lobe and the supramarginal gyrus. Additionally, Cluster 3 responses were stronger for the response-requiring Target in comparison to the Cue. This difference remained consistent across target sides and was evident in both the right and left hemispheres. In addition, this cluster's response-locked activity peaks at the time of the motor response, which also modulates its target-locked activity. Moreover, along the second dimension of the cortical gradient which spans the different modalities of the gradient's periphery, this Cluster's contacts map to the somatomotor periphery and to the high-level core regions of the cortical gradient. Hence, we propose the interpretation that the activity within this cluster is linked to the aspects of motor response planning, rather than motor execution *per se*. We expanded our discussion of this point to clarify it in the revised Discussion (Page 29):

On the other hand, neural activity in Cluster 3 shows sensitivity to stimulus identity, with stronger activation for response-requiring targets than for cues. It is lateralized to the left hemisphere, contralateral to the responding hand, and its response-locked activity peaks at the time of the motor response, which also modulates its target-locked activity. Furthermore, this cluster is anatomically situated between the somatomotor end and transmodal core regions of the core-periphery gradient. Because the patients only responded with their right hand, we cannot completely rule out that the left hemisphere response is simply stronger, and is not related to response planning aspects of the task. However, this cluster contains right

hemisphere contacts as well, and its contacts are localized also in non-motor regions, such as the posterior temporal lobe and supramarginal gyrus. This fact, together with the entire line of evidence mentioned above, support the suggestion that Cluster 3 encodes decisional and/or response aspects.

4. From Discussion:

“Our results suggest that this cluster’s activity represents a key attentional processing, associating perception and action signals.” This is a conclusion statement yet it is vague. It is unclear what key attentional processing is referred to here.

Reply:

Attentional processing refers to the exogenous cueing effect in the long SOA (and in the short SOA) in cluster 2, emerging in the middle of the cluster gradient.

We altered the text accordingly (Page 29):

...our results suggest that the activity in cluster 2 represents a key attentional processing of exogenous cueing effects in both short and long SOAs, associating perception and action signals.

“TPJ contacts were activated also in response to cues, and in congruent trials, when target location corresponded to the location of the preceding cue, therefore suggesting that the TPJ is not just a circuit breaker responding when unexpected and pertinent targets appear and reorienting of attention is needed” – I am not sure I follow this. Cues are salient events that attract exogenous attention independent of whether the target that follows it is congruent with the cue or incongruent.

Reply:

We thank the reviewer for pointing out that this point was not clear enough. We are discussing the role of the TPJ as outlined in the Corbetta and Shulman model based on fMRI data, which postulates that exogenous orienting does not activate the TPJ, which only responds to reorienting to response-relevant stimuli (Corbetta et al., 2008). Corbetta and Shulman propose that when an important stimulus appears outside the current focus of attention, fast-latency signals from the ventral network initiate reorienting by sending a “circuit-breaking” or interrupt signal to dorsal regions, which change the locus of attention (Corbetta and Shulman, 2002). In other words, according to this model, TPJ should not respond to peripheral non-informative cues, only to unexpected (Incongruent) targets. We clarify this point in the revised Discussion (Page 30):

Additionally, our findings concerning the TPJ are not completely consistent with the prominent Corbetta and Shulman model¹⁹. Based on fMRI data, this model postulates that exogenous orienting does not activate the TPJ, which only responds to reorienting to response-relevant targets. Corbetta and Shulman¹⁹ suggest that when an important stimulus appears outside the current focus of attention, fast-latency signals from the ventral network initiate reorienting by sending a “circuit-breaking” or interrupt signal to dorsal regions, which change the locus of attention. Thus, according to this model, TPJ should not respond to peripheral non-informative cues, but only to unexpected incongruent targets. However, we found that TPJ contacts were activated also in response to cues, and also in congruent trials, when target location corresponded to the location of the preceding cue, aligning with previous causal evidence from TMS studies⁷⁶. Therefore, our

*findings suggest that the TPJ is not just a circuit breaker responding when unexpected and pertinent targets appear and reorienting of attention is needed*¹⁹.

Reviewer #2 (Remarks to the Author):

In the present manuscript, the authors study the neurophysiological correlates of exogenous attention using a large dataset of intracranial recordings from the human brain. They employ a well-established classic attention task that has first been reported by Posner et al. in the 1980s.

On the behavioral level, they replicate the finding of inhibition-of-return, albeit only two time points (150ms and 600ms) were probed. On the neural level, they leverage the power of intracranial recordings in humans and extract high-frequency band activity as a proxy of multiunit spiking. Employing a clustering approach, they identify three key clusters spanning sensory, association and motor areas, where only the cluster in the association areas displays a condition-specific difference.

Overall, this manuscript is rather dense in terms of methods, the authors introduce e.g. a multivariate trajectory approach to map condition differences to neural data and employ a variety of data-driven cut-offs to determine the optimal cluster size. Finally, they also incorporate the concept of intrinsic time scales and fiber tracking.

In its present form, the manuscript describes behavior and neural data separately and it remains unclear if the authors truly identified an underlying mechanism that explains the behavioral effect. The manuscript mainly requires several clarifications in terms of methods and might benefit from clearly laying out the overall analytical approach.

Overall, the manuscript seems a bit convoluted, since when ignoring the methods-details, one is left with (a) replication of the behavioral IOR (albeit no facilitation at the early interval), (b) identification/replication (based on functional rather than anatomical data) of the well-known hierarchy sensory->association cortex->motor, (c) the new finding that the IOR is mainly mediated at the level of association areas and (d) that the identified hierarchy tracks the timescale hierarchy (as e.g. identified recently, Gao et al. 2020 eLife). While the paper obviously constitutes a tour-de-force in terms of methods and rather dense, it lacks a clear take-home message of what the key finding is that advances the current conceptual scope. In the title, the authors suggest that the gradient per se might reflect the behavioral correlate, but that seems not substantiated in the manuscript.

Reply:

We appreciate the reviewer's recognition of our manuscript as a "tour de force in terms of methods". We concur with the Reviewer that the manuscript contains numerous analyses and methods aimed at robustly and reliably establishing the neural basis of exogenous attention, which may contribute to its density. We have expanded upon and clarified the theoretical motivations and implications of our work (see Reviewer 1's first comment). In addition, we followed the Reviewer's suggestion and added a paragraph to the

revised Results (page 4) to guide the reader through the analysis and reordered the results to enhance narrative coherence.

The neural analysis approach consisted of several steps. We first aimed to identify contacts with similar temporal activity across all conditions in a data-driven manner, using an adapted clustering trajectory k-means algorithm, which operated on the contacts target-locked temporal responses. We next explored the temporal progression of activity between the identified clusters. Given that the clusters were defined only on the basis of their temporal dynamics, we then investigated the clusters' spatial localization, their white matter connectivity and their spatial relations within the large-scale hierarchy of the cortical gradient, testing the prediction that meaningful clusters will group spatially in an ordered manner. We then turned to characterize how the neural activity across the clusters tracked the visual, attentional and response aspects of the Posner paradigm. Specifically, (1) we tested attentional effects by comparing neural activity across the attention contrasts used for the behavioral analysis; (2) We revealed response-related modulation by examining how differentiating target-locked activity according to the RT affected neural activity; (3) We uncovered visual modulation of neural activity by applying the clustering anew to response-locked activity and studying how separating response-locked activity according to visual stimuli onset time influenced the clusters' neural activity. Finally, (4) we investigated whether the embedding of the cluster gradient in the cortical gradient extended beyond spatial topography and shares a functional hierarchy of temporal integration windows, which could correspond to a proposed theoretical mechanism underlying RT facilitation and IOR^{6,42}.

However, we respectfully disagree with the Reviewer's statement that our manuscript describes behavior and neural data separately and thus the underlying mechanism that explains the behavioral effect remains unclear. Through four different analyses, we have directly linked subjects' behavior (measured as RT in the Posner detection task) with the corresponding neural activity, along the spatiotemporal gradient of clusters that we identified.

First, we identified clusters of contacts based solely on the temporal dynamics of their target-locked neural activity. In three clusters which presented stable activity that differed across the experimental conditions, we sorted the trials according to the RT in individual trials and grouped the trials in 20 RT-bins to gain enough power. This allowed us to perform two analyses, which cross-validated each other:

1. Employing a time-resolved ANOVA, we identified the clusters in which the variance in the target-locked neural activity was significantly explained by the patients' RT. This analysis demonstrated that only in Clusters 2 & 3 did the peak time of the neural activity predict the subsequent behavioral response, across all experimental conditions.
2. Using a cross-correlation analysis, we identified the clusters in which the target-locked neural activity was significantly shifted in time in correlation with the RT. This analysis demonstrated that only in Cluster 2 & 3, the timing of the neural activity followed the subsequent behavioral response, across all experimental conditions.

We then supplemented these analyses by locking the neural activity to the motor response, and identified the clusters of contacts based solely on the temporal dynamics of their response-locked neural activity. Again, in the clusters which corresponded to our previous stable target-locked clusters, we sorted the trials according to the RT in individual trials and grouped the trials in 20 RT-bins to gain enough power. This

allowed us to explore how neural activity is associated with behavior (RT) and perceptual sensitivity. We used the same analyses, cross-validating our results:

3. Using a time-resolved ANOVA, we identified the clusters in which the variance in the response-locked neural activity was significantly explained by the patients' RT. This analysis demonstrated that only in RT-Clusters 2b & 3, the peak time of the neural activity matched the subsequent behavioral response, and that in RT-Clusters 1 & 2a neural activity was explained by visual presentation, across all experimental conditions.
4. Using a cross-correlation analysis, we identified the RT-Clusters in which response-locked neural activity matched the timing of the response. This analysis demonstrated that in RT-Clusters 1 & 2a activity was shifted in time according to the onset of the visual stimulation, whereas only in RT-Cluster 2b & 3 did the timing of the neural activity match the behavioral response, across all experimental conditions.

Collectively, these findings firmly establish a robust and direct link between perception, behavior, and neural activity, across the cluster gradient.

Furthermore, the authors loosely refer to the concept of rhythmic attention, as recently introduced by e.g. Landau et al. or Fiebelkorn and Kastner. This idea remains a bit unevaluated in the paper:

(a) Could it be that the behavioral effect is solely driven by theta-phase dependence (when assuming a 6-7Hz rhythm, then activity at 150ms and 600ms would fall into opposite phase bins, hence, maximizing the behavioral contrast)

Reply:

We thank the Reviewer for this interesting suggestion regarding the potential role of theta-phase in driving the observed behavioral effects. In response, we conducted an extensive analysis to investigate this possibility. To address this hypothesis, we systematically compared the alignment of the instantaneous theta phase at the onset of the Target stimulus (extracted from the raw unfiltered data using a hilbert transform) between conditions with different SOAs and congruence levels. Our analysis involved a mixed ANOVA with repeated-measures Factors of SOA and Congruence, supplemented by a between-subjects factor of Cluster to test if the theta phase effect could arise differentially across different contact clusters. We could not reject the null hypothesis for any of the factors, or their interactions (SOA: $F(1,1348)=0.049$, $p=0.83$; Congruence: $F(1,1348)=0.38$, $p=0.54$; Cluster: $F(6,1348)=0.24$, $p=0.97$; SOA*Cluster: $F(6,1348)=0.26$, $p=0.96$; Congruence*Cluster: $F(6,1348)=0.166$, $p=0.97$; SOA*Congruence: $F(1,1348)=6.17 \cdot 10^{-5}$, $p=0.99$; SOA*Congruence*Cluster: $F(1,1348)=0.33$, $p=0.92$). A Bayesian ANOVA with the same factors (specifying a multivariate Cauchy prior on the effects (van den Bergh et al., 2019) confirmed these negative findings, showing that the null model was the best supported one, with 7.1 (BF_{01}) more evidence for the null compared to the next best model containing the SOA factor. These results suggest that the theta phase cannot explain the behavioral effects, not at the entire sample of contacts and not when looking into particular clusters of contacts. This negative finding is in line with a recent paper, which found no evidence for rhythmic sampling in inhibition of return behavioral effects (Michel & Busch, 2023).

We added this analysis to the revised manuscript's Results section:

The observed differences between SOA and Congruence conditions across clusters could be explained by different theta phases at target onset, as the neural activity at the short-SOA and Long-SOA could fall into opposite phase bins. A control mixed Anova analysis revealed that theta phase could not explain these effects, either across the entire sample of contacts or when looking at particular clusters of contacts (See Supplementary Results). A Bayesian ANOVA with confirmed these negative findings, which are consistent with a recent paper that found no evidence for rhythmic sampling in inhibition of return behavioural effects

60.

This result is also described in more detail in the manuscript's Supplementary Results section:

Theta-phase dependence of neural activity

*To test the hypothesis that the potential role of theta-phase in driving the observed behavioral effects. In response, we conducted an extensive analysis to investigate this possibility. To address this hypothesis, we systematically compared the alignment of the instantaneous theta phase at the onset of the Target stimulus (extracted from the raw unfiltered data using a hilbert transform) between conditions with different SOAs and congruence levels. Our analysis involved a mixed ANOVA with repeated-measures factors of SOA and Congruence, supplemented by a between-subjects factor of Cluster to test if the theta phase effect could arise differentially across different contact clusters. We could not reject the null hypothesis for any of the factors, or their interactions (SOA: $F(1,1348)=0.049$, $p=0.83$; Congruence: $F(1,1348)=0.38$, $p=0.54$; Cluster: $F(6,1348)=0.24$, $p=0.97$; SOA*Cluster: $F(6,1348)=0.26$, $p=0.96$; Congruence*Cluster: $F(6,1348)=0.166$, $p=0.97$; SOA*Congruence: $F(1,1348)=6.17 \times 10^{-5}$, $p=0.99$; SOA*Congruence*Cluster: $F(1,1348)=0.33$, $p=0.92$). A Bayesian ANOVA with the same factors (specifying a multivariate Cauchy prior on the effect*

97 confirmed these negative findings, showing that the null model was the best supported one, with 7.1 (BF01) more evidence for the null compared to the next best model containing the SOA factor. These results suggest that the theta phase cannot explain the behavioral effects, not at the entire sample of contacts and not when looking into particular clusters of contacts.

Finally, we expanded upon this point in the revised Discussion, which now reads:

Although we did not find evidence for the involvement of theta phase in the observed attentional effects, further research is needed to explore the relationship between these phenomena, and test the hypothesis that they interact and influence each other along the attentional and together dynamically contribute to attentional processing.

and (b) the relationship of the temporal hierarchy and the rhythmic sampling (which both constitute some sort of behavioral time signature) should at least be addressed in the discussion to highlight that both phenomena are most likely distinct entities.

Reply:

We thank the Reviewer for this suggestion. The relationship between temporal hierarchy and oscillations is indeed a fundamental question, though still unsolved. A similar gradient of oscillatory frequencies has been shown along the posterior-anterior cortical axis (Mahjoory et al., 2020). Gao and colleagues (2020) suggested that the gradients of oscillations and neural TRWs may (at least in part) share circuit mechanisms at different spatial scales, based on the similarity of these gradients and known mechanisms of

asynchronous and oscillatory population dynamics, analogous to the relationship between characteristic frequency and decay constant in a damped harmonic oscillator model.

In the context of attention, theta rhythms from frontoparietal attentional networks have been proposed to temporally organize sensorimotor functions, creating alternating periods of attentional focus or shift (Fiebelkorn & Kastner, 2019). Thus, conceptually, neural oscillations may serve as 'broadcasted' attentional signals affecting other brain regions. Similarly, TRWs can be thought of as 'receivers' of oscillatory attentional signals, determining how attentional modulation is processed. Future studies should investigate the relationship between these phenomena and test the hypothesis that they interact and influence each other along the attentional gradient, contributing together to attentional processes.

To better address the relationship between temporal hierarchy and the rhythmic sampling, we added the following to the revised Discussion (Page 30):

TRWs may be linked to another neural temporal phenomenon: oscillations. The relationship between the temporal integration hierarchy and oscillations is still unclear. A gradient of oscillatory frequencies, similar to the timescales gradient⁵⁰, has been described along the posterior-anterior cortical axis⁸¹. Gao and colleagues⁵⁰ suggested that the gradients of oscillations and neural receptive windows may (at least in part) share circuit mechanisms at different spatial scales, based on the similarity of these gradients and on known mechanisms of asynchronous and oscillatory population dynamics, analogous to the relationship between characteristic frequency and decay constant in a damped harmonic oscillator model. In the context of attention, theta rhythms from frontoparietal attentional networks have been proposed to rhythmically sample and temporally organize sensorimotor functions, creating alternating periods of attentional focus or shift^{67,70}. Thus, conceptually, neural oscillations may serve as 'broadcasted' attentional signals affecting other brain regions. Similarly, TRWs can be thought of as 'receivers' of oscillatory attentional signals, determining how attentional modulation is processed. For example, the length of the TRW can determine how much of the oscillation's period will be summed together, thus generating a differential modulatory effect of the same oscillation frequency along different parts of the attentional gradient. Although we did not find evidence for the involvement of theta phase in the observed attentional effects, further research is needed to explore the relationship between these phenomena, and test the hypothesis that they interact and influence each other along the attentional gradient, contributing together dynamically to attentional processing.

Taken together, this is an impressive manuscript that is methods-rich and provides several novel ideas how to analyze complex human intracranial data. The link to behavior could be substantiated and the rationale could be better motivated. In addition, several queries remain, mainly to unpack the manuscript and better guide the reader through the various steps and conclusions.

Reply:

We are grateful for the Reviewer's overall positive assessment, and helpful comments. Following their suggestions, we have now added a paragraph and changed the order of the reported results to better walk the reader through the manuscript's narrative.

1. The authors employ a variety of methods, it would be beneficial if the analysis pipeline could be laid out

in the beginning, with a clear rationale why the approaches were chosen. The 8D condition space analysis for instance, on which several subsequent analyses rest, requires a bit more of an illustration and explanation. Ideally, the authors might consider modeling the different possible outcomes to give the reader a clearer intuition of the method. From solely reading the description without any visual depiction, it seems that the 8D space is mainly composed of binary (left or right) values. Furthermore, it remains unclear how these trajectories based on behavioral data were linked to EEG activity.

Reply:

Our apologies for the confusion. The trajectory k-means method takes as input only the neural iEEG time series. The data are then transformed into an 8D matrix, where each dimension corresponds to one of the eight experimental conditions (short/long SOA x congruent/incongruent x contralateral/ipsilateral target relative to the recording contact). The experimental conditions corresponding to the 8D spatial dimensions are the combinations of the 3 experimental factors (SOA, Congruence, Target side), but the values of the matrix are the continuous HFBB power measurements, i.e. the measured neural activity. Thus, the trajectories described in this space consist of the HFBB power measurements in time (see Figure 2A below for an illustration of neural trajectories in a 3D space for ease of visualization, and 2B for an example of neural trajectories of 3 contacts across a 2D experimental conditions space). In other words, the trajectory matrix describes the neural responses in time of all contacts across all experimental conditions simultaneously. The trajectory k-means algorithm then identifies the prototypical temporal dynamics that emerge across groups of contacts (i.e., the clusters). It does this by iteratively minimizing the distances between the contact trajectories and the centroid trajectory of each cluster. Hence, the clusters are determined solely by the temporal dynamics of the iEEG activity.

We now added the illustration and example in Fig. 2 below to the revised Figure S2 and clarify this issue in the revised Methods (page 34):

This yielded 644 contacts (spatial locations are detailed in Table 2 and are best seen in Fig. 2 & S2), and their mean target-locked or response-locked activity time series were transformed into an 8D matrix, where each dimension corresponded to one of the eight experimental conditions (short/long SOA x congruent/incongruent x contralateral/ipsilateral target relative to the recording contact; see Figure S2A-B for an illustration). The trajectories, consisting of the mean target-locked or response-locked HFBB power across the 8-dimensional condition space, were entered into the clustering algorithm. Activity across conditions was z-scored relative to the distribution of the trials' entire duration.

and Results sections (page 4):

This resulted in 644 responsive contacts, for each of which we calculated the temporal trajectory in the 8-dimensional condition space (Congruent / Incongruent Trial X short-SOA / long-SOA X Ipsilateral / contralateral target; see Fig. S2A-B), i.e. the path of each contact's HFBB over time across all experimental conditions.

Figure 2. **(A)** A schematic illustration of two trajectories of contact neural activity in a multi-dimensional experimental condition space (3-dimensional for visualization simplicity). The temporal order of the sampled neural activity, composing the illustrated contact trajectories, is color-coded (red to blue). The dimensions of the 3-D space correspond to the three experimental conditions, such that the trajectories represent the contacts' neural activity measured as HFBB power in all three experimental conditions simultaneously. **(B)** A simplified example transformation of HFBB time series traces of a contact (black) in two experimental conditions into a neural activity trajectory in a 2-dimensional experimental condition space, represented along with the trajectories of two other contacts (yellow & purple). Contact locations in the brain are depicted in the lower left inset (black, yellow & purple circles).

2. In the main clusters, the authors report only very few electrodes (approx. 200 out of a large sample), raising the question what happens at all the other electrodes. Is there truly no effect or modulation? This might also serve as a baseline condition. To me, the link between the three main clusters and the late activation/suppression is unclear. Are these the results of two different analysis or are they the result of the very same analysis?

Reply:

The initial stage of the clustering procedure involved a preliminary separation of contacts into two groups: those that exhibited no response and those that responded in at least one of the experimental conditions, as compared to the baseline distribution (a 200ms window prior to cue presentation). To that end, we used time-resolved t-tests, keeping only the contacts that responded significantly during 100ms ($p \leq 0.05$ uncorrected) or more in at least one of the experimental conditions, yielding 644 responsive contacts out of the 1,403. The reason for this step, which is often used in clustering procedures (see for example Hamilton et al., 2018), was to allow the clustering to mainly separate the different temporal profiles of responses, and not the unresponsive from the responsive trajectories. Notably, clustering the entire sample of 1,403 contacts yielded similar results to that of only the responsive contacts. The trajectories of the 644 contacts were then entered into the k-means algorithm and separated into 6 clusters, in one single analysis. Out of the six resulting clusters, we chose to focus on the three clusters (Clusters 1, 2 and 3) that were the most stable across different k solutions (Figure S3), and that showed variability across the different

experimental conditions (Figure S5). The remaining three clusters (Late activation, Late suppression and Non-responsive) showed a more stable temporal response across all experimental conditions (Figure S2) and their activity was negatively correlated with that of the 3 chosen clusters (Figure S5). Note that the Non responsive cluster contained contacts whose responses were probably idiosyncratic or induced (as opposed to evoked responses) and therefore were averaged out in the Cluster centroid trajectory. The logic that guided our choice to focus on the subset of three clusters was that even if the remaining clusters might contribute to the processing of the different attentional conditions, they cannot explain the differences between them, given that their correlation pattern across experimental conditions is uniform (Figure S5).

We expanded this point in the revised Methods (page 34):

Clusters 1, 2 and 3 were used as functional regions of interest for further analyses. We chose to focus on these clusters because of their stability across clustering solutions and their variable responses across experimental conditions (Figure S5). Even if the remaining clusters might contribute to the processing of the different attentional conditions, they could not explain the differences between them, given that their correlation pattern across experimental conditions was uniform (Figure S5).

In Figure S3 different clusters are marked in color – is this the outcome of a test or did the authors draw these boxes based on visual inspection?

Reply:

We thank the Reviewer for the comment. Not only is this outcome based on visual inspection, but also on statistical tests. A series of contingency tables analyses showed the significant correspondence between the three Clusters we focused on in our work (Clusters 1, 2 and 3) and the color-marked clusters in the figure across different K solutions. Specifically, the main group of contacts composing each of the marked clusters could be significantly mapped to one of these three clusters. For example, in the k=8 solution, 91% of the original Cluster 1 contacts were clustered in cluster #5 (marked in yellow), and represented 98% of the total contact number of this cluster; similarly, 94% of Cluster 2 contacts were assigned to cluster #8 (marked in red), representing 94% of its total contact count, and 79% of Cluster 3 contacts were clustered in cluster #1 (marked in green), comprising 98% of its contacts. This analysis is now reported in the Methods section, and as supplementary Table S1.

3. Figure 2 is labeled as ‘IOR-related neural activity’. Unfortunately, I fail to observe the direct correlate of the behavioral effect with similar activity/behavior at the short interval and a dissociation at the longer interval.

Reply:

We apologize for the lack of clarity. Previous Figure 2 (now revised Figure 3) shows only the long-SOA data, specifically, the neural correlate of the behavioral contrast of Congruent vs. Incongruent targets. If the neural activity is associated with behavior in this cluster then one would expect an earlier neural activity in the Incongruent compared to the Congruent condition in the long-SOA, in coherence with the behavioral effect (IOR: longer RTs in Congruent vs. Incongruent). As discussed above in our reply to the first point of Reviewer 1, regarding the attentional effect in the short-SOA, RT facilitation is an elusive effect, easily masked by other processes (Chica, Martín-Arévalo, et al., 2014; Lupiáñez, 2010; Martín-Arévalo et al., 2016). Our design was not optimal for unmasking the behavioral effect because of the lack of temporal

overlap between cue and target, which is one of the conditions that favors the appearance of facilitation in detection tasks (Chica, Martín-Arévalo, et al., 2014). In addition, the Poffenberger effect we observed further masked the RT facilitation effect. We therefore preferred not to analyze neural data not backed up by significant behavioral effects, as they are much harder to interpret. Yet, current theories postulate that even when difficult to observe behaviorally, the facilitation effect nevertheless exists (Lupiáñez, 2010). Thus, assuming that the facilitation effect is only masked behaviorally, the effect may still be observed neurally. To address reviewers' comments we performed an exploratory analysis on the short-SOA congruence neural effects in Cluster 2 and found a combination of stronger responses for contralateral stimuli (Cue or Target) which are summed together, leading to a faster and stronger activation for contralateral congruent compared to contralateral incongruent cues and targets, and compared to ipsilateral ones (Significant Congruence X Target-side interaction effect, at -60-140ms post target, and main Target-side effect, at 160-300ms, 320-360ms, 440-460ms post target; see the Figure below). The summed neural activity difference indicates a preference for stimuli that repeat in the exact spatial (contralateral) location, aligning with the observed behavioral RT facilitation effect, where RT is quicker for repeated stimuli in a specific location. Therefore, although the behavioral data did not show clear RT facilitation for short SOA congruent trials, the potential neural signature was present. Importantly, both RT facilitation and IOR are classic exogenous attention phenomena, and studying each independently may lead to the identification of the neural basis of exogenous attention.

Because the neural short-SOA effect is much harder to interpret than the IOR neural effect observed in the same cluster, to comply with the Reviewers' comments we decided to cautiously report this analysis in the revised manuscript.

We added this results to the revised Results (page 16):

Despite the lack of a significant behavioral effect of RT facilitation, the effect could be masked by other processes, as is often the case^{6,8,11}. Current theories postulate that even when masked, such facilitation nevertheless exists. We therefore performed an exploratory time-resolved ANOVA analysis with the factors Congruence, Target-side and Hemisphere, to test the attentional neural effect in Cluster 2 also at the short-SOA. In the target time-window, cluster 2 showed a significant Congruence X Target-side interaction effect (-60-140ms post target; largest $p=0.022$) and a main Target-side effect (160-300ms; 320-360ms; 440-460ms post Target onset; largest $p=0.012$; see Fig. S7). This reflects a combination of stronger responses for contralateral stimuli (Cue or Target) which are summed together, leading to a faster and stronger activation for contralateral congruent compared to contralateral incongruent cues and targets, and compared to ipsilateral ones. This differential summed activity translates to a neural preference for stimuli repeating in the same specific spatial (contralateral) location, dovetailing the behavioral RT facilitation effect, in which RT is faster for repeated stimuli at a specific location. Therefore, although the behavioral data did not show clear RT facilitation for short SOA congruent trials, the potential neural signature was present.

We also expanded our discussion of the absence of a behavioral RT facilitation effect (Page 29):

Clusters 2 and 3 are located closer to core regions of the cortical gradient, and overlap with known frontoparietal attention networks^{19,67,68}. The neural activity in Cluster 2, occurring midway along the gradient, is sensitive to cue-target spatial positions and delays, and exhibits IOR-related onset and offset.

Both visual processing of the target and manual response preparation shape the neural activity in this cluster, which is lateralized to the right hemisphere, consistent with lesion and neurostimulation data on IOR^{28–30,33,34}. Despite the fact that we did not find a significant behavioral effect of RT facilitation, the involvement of Cluster 2 neural activity in attentional computation in the short-SOA condition is plausible. First, RT facilitation is an elusive effect, easily masked by other processes^{6,8,11}. Our design was not optimal for unmasking the behavioral effect because of the lack of temporal overlap between cue and target, which is one of the conditions that favors the appearance of RT facilitation in detection tasks⁸. In addition, the Poffenberger effect we observed further masked the RT facilitation effect. Yet, current theories postulate that even when difficult to observe behaviorally, the facilitation effect nevertheless exists^{6,11}. Our exploratory analysis revealed in Cluster 2 at the short-SOA a differential cue-target summed activity, which translates to a neural preference for stimuli repeating in the same specific spatial contralateral location. This neural effect dovetails the behavioral RT facilitation effect, in which RT is faster for repeated stimuli in a specific location. Therefore, our results suggest that this cluster's activity represents key attentional processes for both short and long SOAs, associating perception and action signals.

Figure 1. Exploratory analysis of short-SOA congruence-related neural activity in the target time-window in Cluster 2. Mean target-locked short-SOA activity in Cluster 2 (red), computed over trials pooled across all cluster contacts, for Congruent trials (full lines) and Incongruent trials (dashed lines), when targets were ipsilateral (left) or contralateral to the recording contact (right). Note, that when targets were ipsilateral, Incongruent cues were contralateral (dark blue arrow), and Congruent cues were ipsilateral (light blue arrow), and conversely for contralateral targets. Responses to contralateral cues and targets were stronger than to ipsilateral ones, as shown by a significant Target-side x Congruence effect (shaded light red areas; -60-140ms post Target onset; largest $p=0.022$) and a main Target-side effect (shaded dark red areas; 160-300ms; 320-360ms; 440-460ms post Target onset; largest $p=0.012$). Shaded areas around traces depict SEM; Dashed vertical lines represent Target onset (black) and Cue onset (blue).

Likewise, Figure S7 and S9 seems to show behavioral correlations in all clusters, which is different to their assertions in the main text, which highlights the difference in association cortex.

Reply:

We would like to clarify this point. Figures S7 and S9 show the results of the cross-correlation analysis on the neural activity of Target-locked and Response-locked clusters, respectively. In Figure S7 the analysis demonstrates the association between the timing of target-locked activity and the timing of the subsequent response, only in Clusters 2 and 3, i.e. in the core-end of the cluster gradient. Conversely, the analysis depicted in figure S9 shows the association between the timing of the response-locked activity and that of the visual stimulation, only in RT-Clusters 1 and 2a, i.e. in the peripheral end of the cluster gradient. This was reported in the manuscript, In the Results and Discussion sections, for example on page 27:

Cluster neural activity at one end of the gradient is modulated by visual attributes, while activity at the gradient's other end reflects the timing of the upcoming response, with attentional modulations occurring at the intersection of visual and response signals.

This analysis is conceptually different from that of the attentional contrast (e.g. in Figure 2), which compares all Congruent trials to all Incongruent trials, similarly to the classic contrast used in the behavioral analysis. This direct attentional contrast, which compares target-related activity depending on whether attention was previously congruently captured or not, revealed significant congruence effects only in Cluster 2, as can be clearly seen in Figure 2.

4. Figure 4: Is there any statistical quantification of the fiber tracking? After all it seems that this was obtained in a different dataset.

Reply:

Yes, our fiber tracking analysis used a threshold-free cluster enhancement (TFCE)-based non-parametric t-test with 1000 permutations with an $\alpha=0.05$, to determine the significant tracts connecting the pre and post rolandic contact ROIs. The fiber tracking was indeed performed on a separate dataset: high-resolution 7T MRI scans of 176 healthy individuals of the Human Connectome Project database⁶¹. Thus, the fiber tracking results are statistically defined and not a consequence of clustering, as the Reviewer suggests later in his summary paragraph.

This appears in the Methods section:

Structural connectivity of pre and post rolandic contacts

*To determine the connectional anatomy of the three clusters we used fiber tracking in a sample of 176 healthy controls from the Human Connectome Project database⁶¹ and **used a threshold-free cluster enhancement (TFCE)-based non-parametric t-test to determine the significant tracts.** Contacts of each cluster were fitted with a 3mm radius sphere around them as described above, and labeled as pre or post rolandic, using the central sulcus as a reference point in patients native space (Number of pre and post rolandic*

contacts per cluster: Cluster 1 - 8:60, Cluster 2 - 23:74, Cluster 3 - 34:33). The resulting pre and post rolandic contact spheres were used as region-of-interests (ROIs) to identify white matter fibers connecting them. This fiber-tracking analysis was done on the high-resolution 7T MRI scans of 176 healthy individuals from the Human Connectome Project database ⁶¹ using TrackVis (<http://trackvis.org/>). **The resulting tractography maps were binarized and significant tracts across individuals were determined using a threshold-free cluster enhancement (TFCE)-based non-parametric t-test in FSL (1000 permutations, height threshold of 0.95 to control significance level at $p < 0.05$; <https://fsl.fmrib.ox.ac.uk/fsl/fslwiki/FSL>).**

and in the revised Results (page 13):

We next asked if the contacts within each cluster were structurally connected. We divided each cluster's contacts into pre rolandic contacts, located in the occipital, parietal and temporal lobes, and post rolandic contacts, located in the frontal lobe, using the central sulcus as a landmark. A fiber tracking analysis paired with probability maps in 176 healthy individuals from the Human Connectome Project database ⁵⁷ revealed that white matter tracts significantly connected pre-rolandic and post-rolandic contacts in the three clusters, suggesting these clusters' long-range contacts formed structural networks (Fig. 2D; threshold-free cluster enhancement-based non-parametric t-test, $p < 0.05$).

5. Is the ACW/time scale related to the behavioral effect? Or is this just a secondary observation that is only loosely tied to the main results? In other words, is the change in ACW epiphonemal and solely a result of clustering activity along the hierarchy, or is there something specific about the timescale in association areas that explains the IOR?

Reply:

Our motivation for exploring the temporal window hierarchy was threefold:

1. From a theoretical perspective, we (Seidel Malkinson & Bartolomeo, 2018) and others (Lupiáñez, 2010; Seidel Malkinson & Bartolomeo, 2018) have previously suggested that neural computations of event segregation and integration contribute to the behavioral phenomena of RT-facilitation and IOR (see also Reviewer 1's comment #1). These computations could be a direct result of TRW length.

The integration-segregation hypothesis is now clearly put forth in the revised Introduction (Page 2):

For example, the Cue-target event integration-segregation hypothesis ⁴⁷ postulates that the summation of early and late perceptual processes, spatial selection processes and decision processes, determines together if the net behavioral effect is facilitatory (RT facilitation) or inhibitory (IOR) ^{6,11}. According to this theory, binding together sequential stimuli that share similar features (such as location and close-timing) into a single event file ⁴⁸ can lead to facilitatory effects helping to select the target location in advance ⁶. However, binding can also cause inhibitory effects when the similar sequential stimulus needs to be detected as a new separate event, resulting in a cost in detecting the onset of the target ⁶.

and in the Discussion (Page 31):

Correspondingly, our finding that TRWs lengthen along the cluster gradient reveal potential temporal operations at the basis of exogenous attention. Furthermore, the integration of cue-target responses in Clusters 2 and 3 in the long-SOA could reflect temporal pooling⁵⁹. In Cluster 1, situated lower on the gradient, TRWs are shorter, allowing for segregation of activity even at short delays. In upstream frontoparietal clusters where TRWs are longer, cue- and target-induced responses resulted in a single activity peak. This temporal pooling might group the cue and target in a single event⁸⁰, leading to RT facilitation at short cue-target delays^{6,42,80}. These findings dovetail with the hypothesis that RT facilitation results from a summation of cue-related and target-related responses, thus reflecting hard-wired limitations of the neural system that cannot respond separately to rapidly repeated stimuli, and processes them as a single event^{6,42,80}. Longer cue-target delays could instead provide the system with enough time to segregate cue- and target-related responses^{6,42}.

According to Cue-target event integration-segregation hypothesis⁶ RT facilitation arises when the net effect of facilitatory processes, such as exogenous spatial attention orienting and binding-associated spatial selection benefit, is larger than the detection cost the binding might cause.

2. Conceptually, having established using the contact coordinates on the 2-dimensional cortical gradient reference frame that the attentional clusters are embedded in the cortical gradient and follow a periphery-to-core progression, we wanted to extend this anatomical finding and explore whether the attentional gradient also shares functional features with the cortical gradient. In support of this hypothesis, we show that the length of TRWs increases along the attentional gradient, similarly to the gradual lengthening of timescales across the cortical gradient. This suggests that the embedding of the clusters, which were identified solely based on neural temporal dynamics without any anatomical input, in the cortical gradient is not just anatomical, but also functional.
3. Finally, from a signal processing perspective, our findings that the response to the Cue and Target was segregated in Cluster 1, but integrated in Clusters 2 and 3, is a corollary of the finding that TRW gradually lengthens across the attentional gradient, mechanistically explaining these neural observations in the temporal domain. This finding aligns with our hypothesis about the role of event integration/segregation, by providing a potential computational mechanism, which should be further explored in future research.

Thus, the TRW hierarchy is central to our suggested hypothesis about the neural computations underlying the behavioral phenomena of RT-facilitation and IOR. This temporal hierarchy is not an epiphenomenon of the clustering itself, and mirrors our neural findings in the temporal domain, which are in turn directly associated with behavior through multiple analyses. Still, this question is not completely resolved yet, and our findings lay the foundations for future studies that will further explore the role of the attentional temporal hierarchy in attentional computations.

We expanded this point in the revised Discussion (page 31):

Hence, the TRW hierarchy is central to our proposed hypothesis about the neural computations underlying the behavioral phenomena of RT-facilitation and IOR. This temporal hierarchy is not an epiphenomenon of the clustering per se, and mirrors our neural findings of segregation and integration of neural responses in

the temporal domain, which in turn are directly linked to behavior across multiple analyses. However, the question remains open, and our findings lay the groundwork for future studies to further explore the role of the attentional temporal hierarchy in attentional and other cognitive computations.

6. Later in the manuscript, the authors split cluster 2 into cluster 2a and 2b and the rationale and functional criteria to do so remain unaddressed.

Reply:

The rationale behind this analysis was to explore the visual modulation of the activity across the cluster gradient. To reveal the visual modulation, i.e. how the timing of the target/cue onset affects neural responses and reveal additional variance that can be explained by visual processes, neural activity must be locked to the motor response (the button press), and not to the Target onset, as was done in the main analysis. To avoid biases, we did not use the clusters obtained based on the Target-locked activity, but chose instead to perform the clustering anew on the response-locked trajectories of the 644 responsive contacts. The best k solution was with 7 clusters, and a contingency table analysis allowed us to identify the response-locked clusters that corresponded to Target-locked Clusters 1, 2 and 3. This correspondence analysis showed that when locking the activity to the response the gradient differentiates into 4 clusters: RT-Cluster 1 (46 contacts; 60.3% of target-locked Cluster 1), RT-Cluster 2a (85 contacts; 35.3% of target-locked Cluster 1 and 49.5% Cluster 2), RT-Cluster 2b (79 contacts; 46.4% of target-locked cluster 1 and 31.3% of Cluster 2), and RT-Cluster 3 (39 contacts; 50.7% of target-locked Cluster 3). Thus, the separation of Cluster 2 into two response-locked Clusters (RT-Cluster 2a and RT-Cluster 2b) is based on data-driven clustering and statistical analysis.

This was described in the Results section (Page 17):

We next studied the neural correlates of the visual aspects of the Posner task, by adopting a complementary approach and examining visual modulation of response-locked activity. We applied the trajectory k-means clustering analysis to response-locked activity (Fig. S9 A-C and Movie S2). To map the correspondence of the seven response-locked clusters to the previously identified target-locked clusters, we performed a contingency analysis that revealed four corresponding response-locked clusters ($X^2_{(30)}=1442$; $p < 0.001$; Contingency coefficient 0.83; Fig. 3 and S9D).

To clarify this point we added the following text to the revised Results (Page 19):

To avoid biases, we applied the trajectory k-means clustering analysis to response-locked activity (Fig. S9 A-C and Movie S2) instead of using the clusters obtained from the Target-locked activity.

In sum, this is an interesting manuscript that describes novel electrophysiological data that might explain a well-established behavioral phenomenon. In its present form, the manuscript would benefit from a separate paragraph of overall analysis strategy, that in particular needs to motivate the cluster approach and the link of the 8D behavioral matrix to the EEG analysis. Also, the authors need to highlight if the ACW and fiber tracking analysis is only the consequence of the clustering or if they have evidence for assigning a true functional role. Currently, both are presented as an afterthought. In sum, the manuscript is dense and a bit convoluted, but the overall data and analysis strategy is interesting. Given the multitude of analyses,

evaluating the key statistics is difficult. Including effect sizes for stats would help. Also, based on binomial distributions, the authors might want to assess how many electrodes from the entire sample need to show an effect in order to deem it significant. Currently, some clusters contain very few electrodes, raising the question if this would fall within the expected noise distribution.

Overall, this constitutes an impressive tour-de-force, but given its density the proposed mechanistic link between behavior and large-scale organization remains a bit difficult to understand.

Reply:

We thank the Reviewer for the kind words, and helpful comments. We hope that the revised manuscript is clearer.

Conforming to the Reviewer's request, we edited the manuscript and added a paragraph detailing the overall analysis strategy to the revised Results (page 4).

The neural analysis approach consisted of several steps. We first aimed to identify contacts with similar temporal activity across all conditions in a data-driven manner, using an adapted clustering trajectory k-means algorithm, which operated on the contacts target-locked temporal responses. We next explored the temporal progression of activity between the identified clusters. Given that the clusters were defined only on the basis of their temporal dynamics, we then investigated the clusters' spatial localization, their white matter connectivity and their spatial relations within the large-scale hierarchy of the cortical gradient, testing the prediction that meaningful clusters will group spatially in an ordered manner. We then turned to characterize how the neural activity across the clusters tracked the visual, attentional and response aspects of the Posner paradigm. Specifically, (1) we tested attentional effects by comparing neural activity across the attention contrasts used for the behavioral analysis; (2) We revealed response-related modulation by examining how differentiating target-locked activity according to the RT affected neural activity; (3) We uncovered visual modulation by applying the clustering anew to response-locked activity and studying how separating response-locked activity according to visual stimuli onset time influenced the clusters' neural activity. (4) Finally, we investigated whether the embedding of the cluster gradient in the cortical gradient extends beyond spatial topography and shares a functional hierarchy of temporal integration windows, which could correspond to a proposed theoretical mechanism underlying RT facilitation and IOR^{6,42}.

Regarding the link of the 8D behavioral matrix to the EEG analysis, we addressed the confusion (see Comment #1), and clarified that the clustering is performed on the neural activity matrix itself and not on the behavioral data.

As the Reviewer suggests, in the revised manuscript we highlight the functional relevance of the TRW and fiber tracking analyses.

Revised Results (Page 13):

These results suggest that the functional clusters identified solely on the basis of their temporal responses, correspond to well-defined structural networks.

Following the Reviewer's suggestion, we assessed the probability of finding the observed number of significant contacts that were clustered using the k-means algorithm. We used the binomial distribution to calculate the probability of having 664 contacts out of 1403 showing a significant effect. Using the union

bound, we calculated the maximal success probability per contact, with a 0.05 chance of having a significant result in a time-resolved t-test in at least one of 8 experimental conditions in 5 consecutive time points along the 40 tested time points: $8 \cdot 36 \cdot 0.05^5 = 0.00009$. According to Markov's inequality, for $X \sim \text{Bin}(1403, 0.00009)$ the upper bound probability for finding $X=664$ significant contacts in our sample by chance is < 0.0002 (note that the exact probability is much smaller than this approximation, due to the exponential decay of the binomial distribution). Regarding the partition between the different clusters, a multinomial analysis reveals that the observed distribution of the clusters significantly differed from the expected χ^2 uniform distribution ($\chi^2_{(5)}=267.67$, $p < 0.001$), demonstrating that the cluster partitioning was not due to chance (noise) alone.

We report effect sizes where possible (e.g. Contingency coefficient, page 19). However, we find it infeasible and uninformative to report all effect sizes for the time-resolved ANOVA analyses, which involve dozens of comparisons.

References

- Chica, A. B., Martín-Arévalo, E., Botta, F., & Lupiáñez, J. (2014). The Spatial Orienting paradigm: How to design and interpret spatial attention experiments. *Neuroscience and Biobehavioral Reviews*, *40*, 35–51.
- Corbetta, M., Patel, G., & Shulman, G. L. (2008). The reorienting system of the human brain: from environment to theory of mind. *Neuron*, *58*(3), 306–324.
- Fiebelkorn, I. C., & Kastner, S. (2019). A Rhythmic Theory of Attention. *Trends in Cognitive Sciences*, *23*(2), 87–101.
- Gao, R., van den Brink, R. L., Pfeffer, T., & Voytek, B. (2020). Neuronal timescales are functionally dynamic and shaped by cortical microarchitecture. *eLife*, *9*.
- Hamilton, L. S., Edwards, E., & Chang, E. F. (2018). A Spatial Map of Onset and Sustained Responses to Speech in the Human Superior Temporal Gyrus. *Current Biology: CB*, *28*(12), 1860–1871.e4.
- Lupiáñez, J. (2010). Inhibition of return. *Attention and Time*, 17–34.
- Lupiáñez, J., Klein, R. M., & Bartolomeo, P. (2006). Inhibition of return: Twenty years after. *Cognitive Neuropsychology*, *23*(7), 1003–1014.
- Mahjoory, K., Schoffelen, J.-M., Keitel, A., & Gross, J. (2020). The frequency gradient of human resting-state brain oscillations follows cortical hierarchies. *eLife*, *9*.
- Martín-Arévalo, E., Chica, A. B., & Lupiáñez, J. (2016). No single electrophysiological marker for facilitation and inhibition of return: A review. *Behavioural Brain Research*, *300*, 1–10.
- Michel, R., & Busch, N. A. (2023). No evidence for rhythmic sampling in inhibition of return. *Attention, Perception & Psychophysics*. <https://doi.org/10.3758/s13414-023-02745-x>
- Seidel Malkinson, T., & Bartolomeo, P. (2018). Fronto-parietal organization for response times in inhibition of return: The FORTIOR model. *Cortex; a Journal Devoted to the Study of the Nervous System and Behavior*, *102*, 176–192.
- van den Bergh, D., van Doorn, J., Marsman, M., Draws, T., van Kesteren, E.-J., Derks, K., Dablander, F., Gronau, Q. F., Kucharský, Š., Raj, A., & al., E. (2019). A Tutorial on Conducting and Interpreting a Bayesian ANOVA in JASP. <https://doi.org/10.31234/osf.io/spreb>

REVIEWERS' COMMENTS

Reviewer #1 (Remarks to the Author):

I have to commend the authors for putting in such a thorough effort into the revision and to each and every response to the reviewers. As I indicated in my original review, I do think that this is a tour de force and that there is plenty of interesting and important findings in this manuscript. The same opinion holds after this revision.

I do want to be a bit nit picky, though this in no way undermines the strength of the revision! The authors state that "Both RT facilitation and IOR are classic exogenous attention phenomena, each standing on its own, and studying each one of them, independently of the other, can contribute to establishing the neural basis of exogenous attention" -- while it is true that the neural instantiation of each phenomenon can be studied separately the only way to establish that one effect is an IOR and the other is a facilitation is only by comparing either one to another, or each to a baseline, thus necessitating a measurement of both.

Reviewer #2 (Remarks to the Author):

The authors addressed all queries in detail. This yielded a very dense manuscript with substantial supplemental information, but it lacks a strong take-home message, while providing a very detailed description of the observed effects and their timing using a variety of analytical tools. The abstract remains a bit vague ('provides the scaffold for attention') and the reader is left wondering -how- attention is now supposed to link perception and action. I understand attentional signatures emerges somewhere along the hierarchy and this is well described, but it only remains loosely connected to the the concept of the IOR. Many ideas are implied, but not explicitly spelled out - for example, why do we see the phenomenon of the IOR? Does the hierarchy (which constitutes an inherent biophysical constraint of the network) cause the IOR, or the temporal response window (TRW) predict the timing of the IOR?

However, I recognize that these limitations are discussed and some of these answers will remain speculative and cannot fully be addressed in the current design. I believe the manuscript constitutes a valuable resource for future investigations, since it provides a thorough resource that might inspire multiple lines of follow-up investigation.

A point-by-point response to the reviewers' comments

Reviewer #1's comment: I have to commend the authors for putting in such a thorough effort into the revision and to each and every response to the reviewers. As I indicated in my original review, I do think that this is a tour de force and that there is plenty of interesting and important findings in this manuscript. The same opinion holds after this revision.

I do want to be a bit nit picky, though this in no way undermines the strength of the revision! The authors state that "Both RT facilitation and IOR are classic exogenous attention phenomena, each standing on its own, and studying each one of them, independently of the other, can contribute to establishing the neural basis of exogenous attention" -- while it is true that the neural instantiation of each phenomenon can be studied separately the only way to establish that one effect is an IOR and the other is a facilitation is only by comparing either one to another, or each to a baseline, thus necessitating a measurement of both.

RESPONSE: We appreciate the constructive feedback from the reviewer. We acknowledge their observation regarding the neural basis of the two phenomena. The classical definition of these phenomena relies on the behavioral RT pattern, and while they can be individually measured by comparing RT in previously cued and uncued locations, distinguishing them hinges on the relationship between the RT in these locations. We acknowledge that interpreting their neural substrates poses challenges due to potential overlapping mechanisms.

What we wanted to highlight is that, despite facilitation not reaching statistical significance in our data, consistent with findings in several other studies, Inhibition of Return (IOR) remains a significant exogenous phenomenon with valuable neural bases for exploration. An aspect that may not have been sufficiently emphasized in the initial revision is the novel finding from our study - IOR manifests when a right-lateralized frontoparietal network processes sequential stimuli as separate events sharing the same location. We have now added this bit of text to the abstract:

These findings challenge multi-step models of attention, and suggest that frontoparietal networks, which process sequential stimuli as separate events sharing the same location, drive exogenous attention phenomena such as inhibition of return.

Reviewer #2's comment: The authors addressed all queries in detail. This yielded a very dense manuscript with substantial supplemental information, but it lacks a strong take-home message, while providing a very detailed description of the observed effects and their timing using a variety of analytical tools. The abstract remains a bit vague ('provides the scaffold for attention') and the reader is left wondering -how- attention is now supposed to link perception and action. I understand attentional signatures emerge somewhere along the hierarchy and this is well described, but it only remains loosely connected to the concept of the IOR. Many ideas are implied, but not explicitly spelled out - for example, why do we see the phenomenon of the IOR? Does the hierarchy (which constitutes an inherent

biophysical constraint of the network) cause the IOR, or the temporal response window (TRW) predict the timing of the IOR?

However, I recognize that these limitations are discussed and some of these answers will remain speculative and cannot fully be addressed in the current design. I believe the manuscript constitutes a valuable resource for future investigations, since it provides a thorough resource that might inspire multiple lines of follow-up investigation.

RESPONSE:

To address the reviewer's feedback, we have made several revisions to the manuscript. First, in response to the reviewer's concern, we removed the sentence "provides the scaffold for attention" from the abstract and adjusted the subsequent text. The modified abstract now reads:

Exogenous attention, the process that makes external salient stimuli pop-out of a visual scene, is essential for survival. How attention-capturing events modulate human brain processing remains unclear. Here we show how the psychological construct of exogenous attention gradually emerges over large-scale gradients in the human cortex, by analyzing activity from 1,403 intracortical contacts implanted in 28 individuals, while they performed an exogenous attention task. The timing, location and task-relevance of attentional events defined a spatiotemporal gradient of three neural clusters, which mapped onto cortical gradients and presented a hierarchy of timescales. Visual attributes modulated neural activity at one end of the gradient, while at the other end it reflected the upcoming response timing, with attentional effects occurring at the intersection of visual and response signals. These findings challenge multi-step models of attention, and suggest that frontoparietal networks, which process sequential stimuli as separate events sharing the same location, drive exogenous attention phenomena such as inhibition of return.

In addition, we have refined the concluding paragraph in the Discussion section to enhance clarity. The updated paragraph is as follows:

Our findings challenge traditional attention models of multi-step processing across visual areas. They indicate that exogenous attentional effects follow a continuous neural trajectory across large-scale spatiotemporal gradients, where distinct processes of segregation and integration of attentional events occur. These neural dynamics provide the mechanisms through which the timing of attentional events shape neural processing and consequently our behavior. Our findings suggest that the circuits for attention form a dynamic network, in which attentional effects are properties of the overall network, not separate functions assigned to different parts 89, and thus place exogenous attention processing in the context of the larger topographical organization of the human brain.

We believe these revisions have clarified the key take-home message of our study.

Regarding the role of TRWs in attentional processing, we acknowledge the reviewer's point about the need for further exploration. Nonetheless, our results indicate that this network characteristic drives the

emergence of Inhibition of Return (IOR). To provide additional clarification, we have revised the Discussion section, which now reads:

Hence, our results contribute to resolving the longstanding debate surrounding the nature of IOR. In Clusters 2 and 3, IOR was linked to a segregation of neural responses, with distinct peaks corresponding to cues and targets. Notably, in Cluster 2 (encompassing the angular gyrus and lateral prefrontal cortex), the timing of these distinct peaks, as well as their decay, mirrored behavioral IOR. Consequently, our findings provide a refined anatomical and functional specification of earlier results obtained from studies involving brain-damaged patients 33,83 and those employing transcranial magnetic stimulation (TMS) on the parietal cortex 28,29,84. This more detailed insight contributes to a better understanding of the precise temporal mechanisms underpinning cognitive processes.